

# Artificial event horizons in Weyl semimetal heterostructures and their non-equilibrium signatures

**Christophe De Beule[1]\*, Solofo Groenendijk[1], Tobias Meng[2] and Thomas L. Schmidt[1]**

**1** Department of Physics and Materials Science,
University of Luxembourg, L-1511 Luxembourg, Luxembourg
**2** Institute for Theoretical Physics and Würzburg-Dresden Cluster of
Excellence ct.qmat, Technische Universität Dresden, 01069 Dresden, Germany

⋆ christophe.debeule@uni.lu

## Abstract

We investigate transport in type-I/type-II Weyl semimetal heterostructures that realize effective black- or white-hole event horizons. We provide an exact solution to the scattering problem at normal incidence and low energies, both for a sharp and a slowly-varying Weyl cone tilt profile. In the latter case, we find two channels with transmission amplitudes analog to those of Hawking radiation. Whereas the Hawking-like signatures of these two channels cancel in equilibrium, we demonstrate that one can favor the contribution of either channel using a non-equilibrium state, either by irradiating the type-II region or by coupling it to a magnetic lead. This in turn gives rise to a peak in the two-terminal differential conductance which can serve as an experimental indicator of the artificial event horizon.



# 1   Introduction

Hawking radiation is the phenomenon whereby black holes slowly evaporate by emitting thermal radiation due to quantum fluctuations near the event horizon [1, 2]. It is one of the most exotic predictions of quantum field theory in a curved spacetime but its experimental verification remains elusive. Indeed, the corresponding Hawking temperature is inversely proportional to the mass of the black hole such that the effect is masked by the cosmic microwave background for generic black holes. However, analogs of Hawking radiation can arise in other physical systems that are more amenable to experimental verification, featuring artificial event horizons sharing many similarities to their gravitational counterpart [3]. This field was pioneered by Unruh who proposed such an analog at the interface between subsonic and supersonic flow in a hydrodynamic system [4]. In condensed-matter physics, similar black-hole analogs have been proposed in Bose-Einstein condensates [5,6], optical systems [7], borophene [8], one-dimensional fermionic chains [9], among others, and recently in Weyl semimetals [10–16]. In this work, we study electronic analogs of stimulated Hawking emission in heterostructures containing an interface of a type-I and type-II Weyl semimetal, as illustrated in Fig. 1(a). To the best of our knowledge, our work provides the first explicit calculation of physical observables in Weyl semimetal black hole analogs, and does so using a minimal model that captures all salient features of Weyl semimetals.

Weyl semimetals host quasiparticles near generic crossings of the energy bands whose low-energy physics are captured by a Weyl Hamiltonian ($\hbar = v_F = 1$) [17]

$$\mathcal{H}_\chi = E_\chi + V\left(k_z - k_{\chi z}\right) + \chi\left(\boldsymbol{k} - \boldsymbol{k}_\chi\right)\cdot\boldsymbol{\sigma}, \tag{1}$$

where $\boldsymbol{k} = (k_x, k_y, k_z)$ and $\left(E_\chi, \boldsymbol{k}_\chi\right)$ is the energy and position of the Weyl node in the Brillouin zone and $\boldsymbol{\sigma} = \left(\sigma_x, \sigma_y, \sigma_z\right)$ is the vector of Pauli matrices. Weyl nodes carry a net Berry flux $\chi = \pm 1$ and necessarily come in pairs as the total Berry flux in the Brillouin zone vanishes [18]. Depending on the tilt, given by the second term in Eq. (1), one can distinguish two types of Weyl semimetals based on the Fermi surface topology, as illustrated in Fig. 1(b). When $V^2 < 1$, the Fermi surface at the Weyl node is a point (type I). At the critical tilt $V^2 = 1$, there is a Lifshitz transition and the system is a nodal line semimetal. For $V^2 > 1$, the nodal line evolves into an electron and hole pocket that touch at the Weyl node (type II), shown in Fig. 1(b) and (c) [19].

The connection between type-I/type-II heterostructures and Weyl fermions in an effective curved spacetime is made explicit by writing down the covariant form of the Weyl equation in a general spacetime with the tetrad formalism [10, 20, 21],

$$\sigma^a e_a^\mu\left(\partial_\mu + \Omega_\mu\right)\psi = 0, \tag{2}$$

where we use the Einstein summation convention with $\mu = 0, 1, 2, 3$ corresponding to general spacetime coordinates and $a = 0, 1, 2, 3$ to local inertial coordinates and where $e^\mu_a$ are tetrad components. A short introduction to the tetrad formalism and more details on the covariant Weyl equation are given in App. A. Here, we also introduced $\sigma^a$ given by the identity and the Pauli matrices $\chi\boldsymbol{\sigma}$, and the spin connection $\Omega_\mu$. The spin connection ensures covariance since $\partial_\mu\psi$ does not transform as a spinor under local Lorentz transformations. The corresponding Weyl Hamiltonian can be written as $\mathcal{H} = \boldsymbol{\alpha} \cdot \boldsymbol{k} - i\Upsilon$ with

$$\alpha^i(p) = (e^0_{\ b}\sigma^b)^{-1}e^i_{\ a}\sigma^a, \qquad \Upsilon(p) = (e^0_{\ b}\sigma^b)^{-1}e^\mu_{\ a}\sigma^a\Omega_\mu, \tag{3}$$

where $p$ is a point in spacetime. If we compare this to Eq. (1), we identify

$$e^\mu_{\ a}(z) = \delta^\mu_a + V(z)\delta^\mu_3\delta^0_a, \tag{4}$$

which correspond to the so-called acoustic metric [4]:

$$g_{\mu\nu} = \begin{pmatrix} V^2-1 & 0 & 0 & -V \\ 0 & 1 & 0 & 0 \\ 0 & 0 & 1 & 0 \\ -V & 0 & 0 & 1 \end{pmatrix}, \tag{5}$$

and which follows from $g^{\mu\nu} = e^\mu_{\ a}e^\nu_{\ b}\eta_{ab}$ with $\eta_{ab} = \mathrm{diag}\,(-1, 1, 1, 1)$ the Minkowski metric of flat spacetime (see App. A). The spin connection becomes

$$\Omega_\mu = \frac{\chi\sigma^3}{2}\left(\delta^3_\mu - V\delta^0_\mu\right)V', \tag{6}$$

with $V' = \partial V/\partial z$, and the corresponding Weyl equation can then be written as

$$i\partial_t\phi = \left[-i\left(\chi\boldsymbol{\sigma} + V\boldsymbol{e}_z\right)\cdot\nabla - iV'/2\right]\phi \equiv \mathcal{H}_\chi\phi, \tag{7}$$

where the extra term proportional to $V'$ comes from the spin connection and ensures that the Hamiltonian is Hermitian for a position-dependent tilt [14].

We can thus interpret a tilted Weyl cone in terms of a free Weyl fermion in an effective curved spacetime with line element

$$ds^2 = g_{\mu\nu}dx^\mu dx^\nu = -dt^2 + [dz - V(z)dt]^2 + dx^2 + dy^2, \tag{8}$$

whose null trajectories at normal incidence are given by

$$\frac{dx}{dt} = \frac{dy}{dt} = 0, \qquad \frac{dz}{dt} = V(z) \pm 1, \tag{9}$$

where $\pm$ corresponds to so-called copropagating and counterpropagating solutions, respectively. We note that one obtains the same equations from the semiclassical equations of motion [22]

$$\dot{\boldsymbol{r}} = \frac{\partial E}{\partial\boldsymbol{k}} - \chi\dot{\boldsymbol{k}}\times\boldsymbol{\Omega}, \qquad \dot{\boldsymbol{k}} = -\frac{\partial V}{\partial z}k_z\boldsymbol{e}_z, \tag{10}$$

in the absence of external electromagnetic fields and where $\boldsymbol{\Omega}(\boldsymbol{k}) = \boldsymbol{k}/2k^3$ is the Berry curvature and $E(\boldsymbol{k}) = Vk_z \pm |\boldsymbol{k}|$. One can check that at normal incidence ($k_x = k_y = 0$) the first equation of motion yields Eqs. (9).

Now consider the case where $V(z)$ is monotonic and $|V(0)| = 1$, corresponding to an interface at $z = 0$ between a type-I and type-II Weyl semimetal [Fig. 1(a)]. While the type-I region ($|V| < 1$) supports trajectories that propagate in both directions, the type-II region ($|V| > 1$)

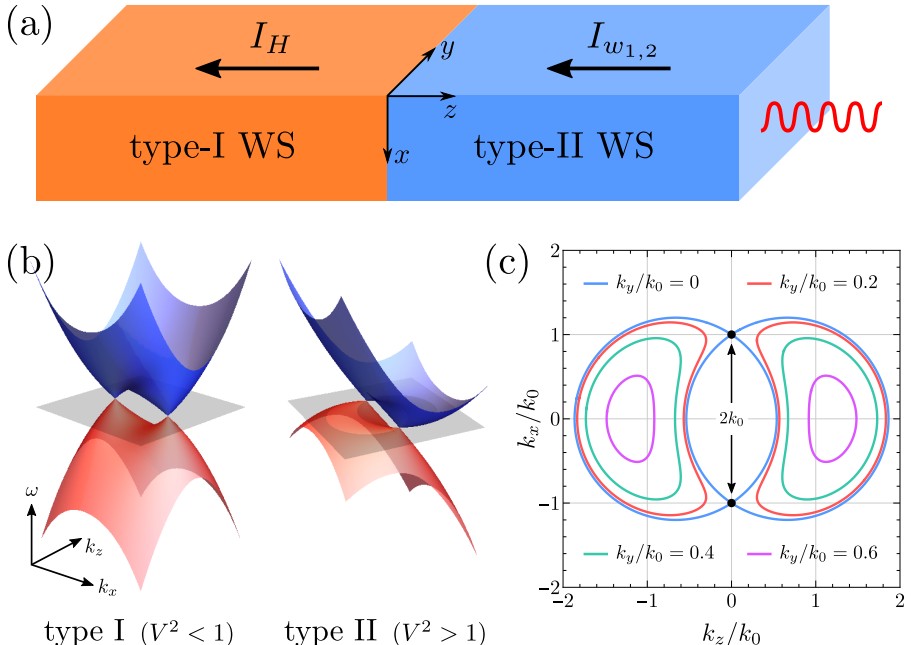

Figure 1: (a) Weyl semimetal (WS) heterostructure where the type-II region is irradiated, which gives rise to a Hawking current between the type-II and type-I region. (b) Energy dispersion for $k_y = 0$. (c) Zero-energy Fermi surface in the type-II phase for $V/v_F = 1.2$ where the Weyl nodes are shown as black dots.

only supports unidirectional trajectories. Semiclassically, the interface gives rise to a turning point where the group velocity of counterpropagating modes vanishes. Hence, we can regard the interface as an artificial event horizon, where the type-I and type-II regions respectively correspond to a normal region of spacetime and a "black hole" ($V > 1$) or "white hole" ($V < -1$) spacetime. However, we note that the acoustic metric is not a solution of the Einstein equations in four-dimensional spacetime. Actually, since the effective spacetime is essentially two dimensional, it is conformally flat [3]. However, due to the horizon, the effective spacetime is not equivalent to a flat spacetime globally. For these reasons, the acoustic metric that we consider is not equivalent to a black hole from general relativity. Instead, the correspondence resides in the fact that both feature an event horizon. Hence, the effective spacetime has a similar causal structure as that of a real black hole. We note that an exact mapping between a type-I/type-II interface to a Schwarzschild black hole in Gullstrand-Painlevé coordinates is obtained for a radial tilt profile $V(r) = -\sqrt{r_S/r}$ with $r_S$ the Schwarzschild radius [10].

In this work, we investigate transport through a type-I/type-II interface using a minimal model for a Weyl semimetal with a tilt profile $V(z)$. We consider both fast and slow varying tilt profiles relative to the Fermi wavelength. In both cases, we obtain low-energy expressions for the scattering matrix and the tunneling rates at normal incidence. In the case of a slowly-varying tilt profile with a linear horizon, for energies $\omega$ close to the Weyl node, counterpropagating particles tunnel through the effective horizon from inside the black hole region via two channels with probability

$$T_{1,2} = \frac{1}{1 + e^{\pm 2\pi\omega/\hbar V'(0)}}, \tag{11}$$

where $cV'(0)$ is the effective gravitational field strength at the horizon with $c$ the speed of light [1–3]. In equilibrium, both channels contribute equally and since $T_1 + T_2 = 1$ there is no net analog of Hawking radiation [10,16]. We therefore propose a means of creating a stationary non-equilibrium distribution by irradiating the type-II region with light or by injecting

a spin-polarized current from a magnetic lead. Both cases favor the occupation of one of the two channels, yielding a net non-equilibrium Hawking effect. Summing over all transverse channels, we then find that the differential conductance is asymmetric about the energy of the Weyl node and features a peak whose position and height is characterized by the slope of the tilt profile at the horizon.

This paper is organized as follows: In Sec. 2, we introduce the continuum model for the Weyl semimetal heterostructure and in Sec. 3 we solve the scattering problem at normal incidence for the case of a fast or slow varying tilt profile. In the former case, we employ standard scattering theory where the horizon only enters through the boundary conditions, while in the latter we use the WKB approximation in combination with an approximate solution that is valid close to a linear horizon. In Sec. 4, we discuss how to obtain a net Hawking effect out of equilibrium. In particular, we show how to favor the occupation of one of the two counterpropagating modes that tunnel across the horizon, and we calculate the differential conductance. Finally, we present our conclusions in Sec. 5.

## 2 Model

We start from a minimal model for a tilted Weyl semimetal with two isotropic Weyl cones that are cotilted normal to the axis along which the nodes lie [23]. For a bulk system, the Hamiltonian is given by $\hat{H} = \sum_{\boldsymbol{k}} \hat{c}_{\boldsymbol{k}}^\dagger \mathcal{H}(\boldsymbol{k}) \hat{c}_{\boldsymbol{k}}$ with $\hat{c}_{\boldsymbol{k}} = (\hat{c}_{1\boldsymbol{k}}, \hat{c}_{2\boldsymbol{k}})^t$ and

$$\mathcal{H}(\boldsymbol{k}) = V k_z \sigma_0 + \frac{1}{2k_0}\left(|\boldsymbol{k}|^2 - k_0^2\right)\sigma_x + k_y \sigma_y + k_z \sigma_z, \tag{12}$$

with $\boldsymbol{k} = (k_x, k_y, k_z)$ and $\sigma_0$ the identity matrix. The Weyl nodes with chirality $\chi = \pm 1$ are located at momenta $\pm k_0 \boldsymbol{e_x}$ and we set $v_F = 1$. The tilt $V$ is applied along the $z$ axis and given by the first term of Eq. (12). In general, our model only has a mirror symmetry about the $yz$ plane: $\mathcal{H}(-k_x, k_y, k_z) = \mathcal{H}(k_x, k_y, k_z)$ and a chiral symmetry given by $\sigma_z \mathcal{H}(k_x, k_y, -k_z)\sigma_z = -\mathcal{H}(k_x, k_y, k_z)$.

For $|V| < 1$, this model gives a type-I Weyl semimetal with a point Fermi surface at the Weyl nodes, while for $|V| > 1$ we obtain a type-II Weyl semimetal. This is illustrated in Fig. 1(b), where we show the energy dispersion relation for $k_y = 0$ in both phases. In the type-II phase, the zero-energy Fermi surface consists of electron and hole pockets touching at the Weyl nodes. The connectivity of these pockets depends on the details of the tilting term. For our model, the pockets form a crescent between the Weyl nodes, as shown in Fig. 1(c). For example, isolated pairs of electron and hole pockets are obtained from the tilting term $\propto k_x k_z$ which tilts the Weyl cones in opposite directions and preserves inversion symmetry. Note also that the second-order terms in Eq. (12) break Lorentz covariance and introduce a length scale $k_0^{-1}$ which regularizes divergences in the limit $|V| \to 1$, known as the trans-Planckian problem [2, 3]. Moreover, these terms keep the Fermi surface finite in the overtilted regime and therefore they cannot be neglected in a realistic system, see Fig. 1(c).

We now consider an interface between a type-I and a type-II Weyl semimetal modeled by a tilt profile $V(z)$ with $V(+\infty) = V_R$ and $V(-\infty) = V_L$ such that $|V_L| < 1$ (type I) and $|V_R| > 1$ (type II) as shown in Fig. 1(a). Since translation symmetry is only broken along the tilt axis, the single-particle wave function can be written as $\Psi(\boldsymbol{r}, t) = e^{i(\boldsymbol{k}_\perp \cdot \boldsymbol{r}_\perp - \omega t)}\phi(z)$ with $\boldsymbol{r}_\perp = (x, y)$, where $\boldsymbol{k}_\perp = (k_x, k_y)$ and $\omega$ are the conserved transverse momentum and energy, respectively. The continuum Hamiltonian becomes

$$\hat{H} = \sum_{\boldsymbol{k}_\perp} \int_{-L/2}^{L/2} dz \, \hat{\psi}_{\boldsymbol{k}_\perp}^\dagger(z) \mathcal{H}(\boldsymbol{k}_\perp, -i\partial_z) \hat{\psi}_{\boldsymbol{k}_\perp}(z), \tag{13}$$

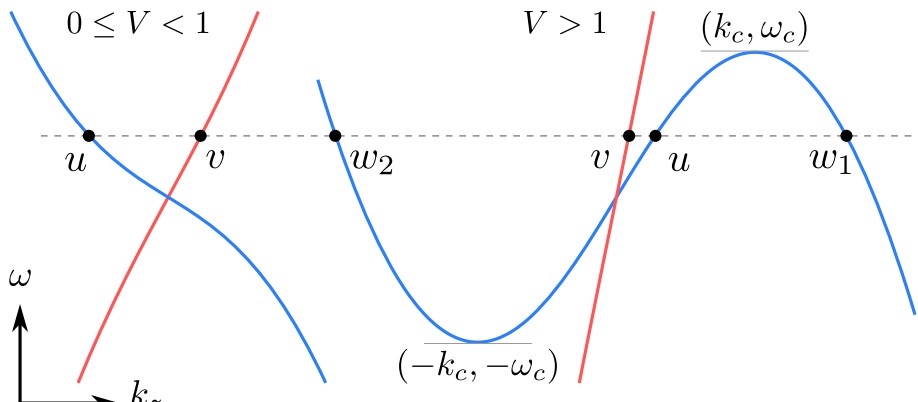

Figure 2: Dispersion relation for $k_\perp = (\pm k_0, 0)$ in the undertilted $(0 \leq V < 1)$ and overtilted case $(V > 1)$ where the copropagating and counterpropagating branch correspond to the red and blue solid lines, respectively.

with $L$ the system length, $\hat{\psi}_{k_\perp}(z) = L^{-1/2} \sum_{k_z} \hat{c}_k e^{ik_z z}$, and

$$\mathcal{H} = \frac{1}{2i} \frac{\partial V}{\partial z} \sigma_0 - i [V(z)\sigma_0 + \sigma_z] \partial_z - \partial_z^2 \sigma_x + \left(k_\perp^2 - 1/4\right) \sigma_x + k_y \sigma_y, \tag{14}$$

in dimensionless form such that now $\mathcal{H}$ is given in units of $2\hbar v_F k_0$, $z$ in units of $1/2k_0$, momenta in units of $2k_0$, and the tilt $V$ in units of $v_F$ such that the transition between the type-I and type-II phase corresponds to $V^2 = 1$. Similarly as before, the first term in the Hamiltonian (14) ensures Hermiticity. We now first consider the special case $k_\perp = (\pm 1/2, 0)$ which we call the *Hawking channel*. Later we return to the full problem which we solve numerically using the KWANT Python package [24].

One can show that any solution $\psi(z, t)$ of the wave equation $\mathcal{H}_0 \psi = i\partial_t \psi$, where $\mathcal{H}_0 = \mathcal{H}(\pm 1/2, 0, -i\partial_z)$, obeys a continuity equation

$$\partial_t(\psi^\dagger \psi) + \partial_z j = 0, \tag{15}$$

with current density

$$j(z, t) = \psi^\dagger [V(z)\sigma_0 + \sigma_z] \psi + 2 \operatorname{Im}\left(\psi^\dagger \sigma_x \partial_z \psi\right), \tag{16}$$

which is constant in space and time for stationary states $\psi(z, t) = e^{-i\omega t} \phi(z)$ as our problem is effectively one dimensional. For constant tilt, the eigenstates of $\mathcal{H}_0$ are given by plane waves, $\phi(z) = e^{ikz} \varphi_{k\lambda}$, with $\lambda = \pm$ and

$$\varphi_{k+} = \frac{1}{\sqrt{2c(1+c)}} \begin{pmatrix} 1+c \\ k \end{pmatrix}, \tag{17}$$

$$\varphi_{k-} = \frac{1}{\sqrt{2c(1+c)}} \begin{pmatrix} -k \\ 1+c \end{pmatrix}, \tag{18}$$

where $c(k) = \sqrt{1+k^2}$. Note that we dropped the subscript $z$ for $k_z$ and that $\{\mathcal{H}_0, \sigma_y\} = 0$ such that $\varphi_{k-} = -i\sigma_y \varphi_{k+}$. For concreteness, we now take $V \geq 0$ in the remainder of this work. In this case, the $\lambda = +$ ($\lambda = -$) branch is referred to as the copropagating (counterpropagating) branch [3]. This nomenclature follows from the dispersion relations

$$\omega = k[V + \lambda c(k)]. \tag{19}$$

Note that the same dispersion is also obtained for a Bose-Einstein condensate with macroscopic velocity $V$ and contact interactions in mean-field approximation [5]. Here, we have one copropagating mode and three counterpropagating modes for a given energy, see Fig. 2. Closed-form expressions for the momenta of these modes exist, but they are unwieldy and not very insightful. Up to first order in $\omega$, we find

$$k_v = \frac{\omega}{V+1} + \mathcal{O}(\omega^3), \tag{20}$$

$$k_u = \frac{\omega}{V-1} + \mathcal{O}(\omega^3), \tag{21}$$

$$k_{w_{1,2}} = \pm\sqrt{V^2-1} + \frac{V\omega}{1-V^2} + \mathcal{O}(\omega^2), \tag{22}$$

where co- and counterpropagating modes are labeled by $v$ and $\{u, w_1, w_2\}$, respectively, and the corresponding spinors are given in lowest order by

$$\varphi_v \simeq \left(1, \frac{\omega}{2(1+V)}\right)^t, \tag{23}$$

$$\varphi_u \simeq \left(\frac{\omega}{2(1-V)}, 1\right)^t, \tag{24}$$

$$\varphi_{w_{1,2}} \simeq \frac{1}{\sqrt{2V}}\left(\mp\sqrt{V-1}, \sqrt{V+1}\right)^t, \tag{25}$$

which are normalized up to first order in $\omega$. It follows that the $w$ modes are evanescent for $0 \le V < 1$, while for $V > 1$ all counterpropagating modes are scattering states for

$$|\omega| < \omega_c = k_c\left[V - c(k_c)\right], \tag{26}$$

with $k_c = (V^2 - 4 + V\sqrt{V^2+8})^{1/2}/(2\sqrt{2})$ [5]. Here, the group velocity of the counterpropagating modes vanishes, corresponding to a classical turning point (Fig. 2). The low-energy expansions of the wavevectors and spinors are valid away from these extrema.

## 3 Scattering at an effective horizon

In this section, we solve the scattering problem for the Hawking channel analytically at low energies in two limits, namely, when the tilt profile $V(z)$ varies fast or slow compared to the Fermi wave length. In the former case, we use plane-wave scattering modes together with a boundary condition that conserves the current. In the latter case, we calculate the WKB wave function away from the classical turning point in combination with an approximate solution near a linear horizon valid at low energies close to the Weyl node, in order to match the WKB wave functions at either side of the horizon.

### 3.1 Slowly-varying tilt profile

#### 3.1.1 WKB solution

We first consider the slowly-varying limit for which the tilt profile is slowly varying on the scale of the Fermi wavelength. To this end, we use a WKB *ansatz*

$$\phi(z) = e^{i\int^z dz' k(z')}\varphi(z), \tag{27}$$

with [25]

$$\varphi(z) = a(z)\varphi_{k+} + b(z)\varphi_{k-}, \tag{28}$$

where $k(z)$, $a(z)$, and $b(z)$ are to be determined from the wave equation. Here, the spinors $\varphi_{k\pm}$ are eigenstates for constant tilt [Eqs. (17) and (18)], in which case $k$ corresponds to the momentum in the $z$ direction. If we plug the *ansatz* in the wave equation, we obtain

$$-i\left[k\left(V\sigma_0+\sigma_z\right)+k^2\sigma_x-\omega\right]\varphi = \frac{1}{2}\left(V'\sigma_0+2k'\sigma_x\right)\varphi \\ +\left(V\sigma_0+\sigma_z+2k\sigma_x\right)\varphi'-i\sigma_x\varphi'', \tag{29}$$

where primes indicate derivatives with respect to $z$. So far, everything is exact. We now make a WKB approximation for a slowly-varying tilt profile by only keeping terms up to first order in $V'$ and dropping all terms proportional to $\left(V'\right)^2$ and $V''$. Next, we multiply with the spinor $\left(\varphi_{k\pm}\right)^t$ from the left which yields two coupled equations for the spinor coefficients $a(z)$ and $b(z)$,

$$-ia\left[k\left(V+c\right)-\omega\right] = \frac{av'_+}{2}+a'v_+ +\left(\frac{1}{c}-V\right)\frac{bk'}{2c^2}+\frac{b'k}{c}, \tag{30}$$

$$-ib\left[k\left(V-c\right)-\omega\right] = \frac{bv'_-}{2}+b'v_- +\left(\frac{1}{c}+V\right)\frac{ak'}{2c^2}+\frac{a'k}{c}, \tag{31}$$

where

$$v_\pm(k) = V\pm\frac{d}{dk}kc(k) = V\pm\left(2c-\frac{1}{c}\right), \tag{32}$$

is the group velocity where we used $dc/dk=k/c$.

In lowest order, we have $a=0$ ($b=0$) for counterpropagating (copropagating) modes since this corresponds to the case of constant tilt. Furthermore, any terms containing derivatives of the tilt will be small. Hence, we find $\omega = k(z)[V(z)+\lambda c(k(z))]$ where $k(z)$ is the semiclassical momentum. We now assume that first-order corrections to $a$ ($b$) for counterpropagating (copropagating) modes are proportional to $V'$. For example, for counterpropagating modes ($\lambda=-1$), Eqs. (30) and (31) become

$$2ikca = \left(V-\frac{1}{c}\right)\frac{bk'}{2c^2}-\frac{b'k}{c}, \tag{33}$$

$$\frac{b'}{b} = -\frac{v'_-}{2v_-}, \tag{34}$$

where we dropped higher-order terms taking into account $a\propto V'$ for counterpropagating modes. The second equation is solved by $b(z)\propto 1/\sqrt{v_-(z)}$ and plugging this back into the first equation yields

$$\frac{ia}{b} = \frac{1}{4c^2}\left[\left(V-\frac{1}{c}\right)\frac{k'}{kc}+\frac{v'_-}{v_-}\right]. \tag{35}$$

Up to first order, the WKB solution of the counterpropagating branch thus becomes

$$\phi_\mu(z) = c_\mu\frac{e^{i\int^z dz'k_\mu(z')}}{\sqrt{v_\mu(z)}}\left(\sigma_0+\frac{ia}{b}\sigma_y\right)\varphi_\mu(z), \tag{36}$$

where $c_\mu$ is a constant and $\mu\in\{u,w_1,w_2\}$. Here $k_\mu(z)$ is a solution of the local dispersion relation, $v_\mu(z)=v_-(k_\mu(z))$ is the corresponding group velocity defined in Eq. (32), and $\varphi_\mu(z)=\varphi_{k_\mu(z)-}$ is the wavefunction, given in Eq. (18). The WKB solution breaks down at the classical turning point where the group velocity vanishes, which is located across the horizon ($V>1$) at finite energies [Eq. (26)]. The group velocity of copropagating ($v$) modes never vanishes such that the WKB solution is valid everywhere and the $v$ modes are therefore

perfectly transmitted in the slowly-varying limit. Indeed, in lowest order an incident $v$ mode has nowhere else to go since it is decoupled from the $u$ and $w$ modes. Observe also that the first-order correction in Eq. (36) couples the $\{u, w_1, w_2\}$ and $v$ modes since $\varphi_{k+} = i\sigma_y \varphi_{k-}$.

In the following, we are mostly interested in the lowest-order result, where the copropagating and counterpropagating branches are decoupled. From Eq. (35), we find this generally holds for

$$\left| k'/k \right|, \; \left| v'/v \right| \ll 1, \tag{37}$$

which by definition is satisfied for a slowly-varying tilt profile away from the turning point.

We now consider a slowly-varying tilt profile with $V(-\infty) = V_L$ with $0 \leq V_L < 1$ and $V(+\infty) = V_R > 1$ that increases monotonically, such that the WKB solutions are valid sufficiently far away from the horizon, where they eventually reduce to the solutions for constant tilt. To solve the scattering problem for the counterpropagating modes, we need to connect wave functions on opposite sides of the horizon. Hence, we need to find a solution that is valid close to the horizon and match it to the WKB solution in a region where both solutions hold simultaneously.

### 3.1.2 Solution near a linear horizon

Let us place the horizon at $z = 0$ such that $V(0) = 1$. We further restrict ourselves to a linear horizon, i.e., close to the origin we assume the tilt profile can be approximated as $V(z) = 1 + \alpha z$ with $\alpha > 0$. In this case, the classical turning point is located at $z_c = 3|\omega|^{2/3}/2\alpha$. In the linear regime, the wave equation becomes

$$\frac{1}{2}(\alpha - 2i\omega)\phi + \begin{pmatrix} 2 + \alpha z & 0 \\ 0 & \alpha z \end{pmatrix} \phi' - i\sigma_x \phi'' = 0, \tag{38}$$

or explicitly

$$\frac{1}{2}(\alpha - 2i\omega)\phi_1 + (2 + \alpha z)\phi_1' - i\phi_2'' = 0, \tag{39}$$

$$\frac{1}{2}(\alpha - 2i\omega)\phi_2 + \alpha z\phi_2' - i\phi_1'' = 0. \tag{40}$$

This yields a fourth-order linear differential equation for $\phi_1$ or $\phi_2$. If we further assume that $|\alpha z| \ll 1$ and $\alpha, |\omega| \ll 1$, we find

$$\phi_2^{(4)} + 2\alpha z \phi_2^{(2)} + (3\alpha - 2i\omega)\phi_2^{(1)} \simeq 0, \tag{41}$$

and $\phi_1 \simeq (i/2)d\phi_2/dz$. Equation (41) can be solved exactly, giving

$$\phi_2(z) = c_0 + \sum_{n=1}^{3} c_n z^{n-1} {}_1F_2\left(a_n; b_n; \frac{-2\alpha z^2}{9}\right), \tag{42}$$

where ${}_1F_2$ is a generalized hypergeometric function with $a_n = (2n - 1)/6 - i\omega/3\alpha$, $b_1 = \left\{\frac{1}{3}, \frac{2}{3}\right\}$, $b_2 = \left\{\frac{2}{3}, \frac{4}{3}\right\}$, and $b_3 = \left\{\frac{4}{3}, \frac{5}{3}\right\}$, and where $c_0$ and $c_n$ are constants.

### 3.1.3 Connection formulas and S matrix

To determine the $S$ matrix, we have to match the asymptotic forms of Eq. (42) to the WKB solutions. To simplify this calculation, we choose a particularly convenient solution that is purely evanescent outside of the horizon ($z < 0$) [26, 27]. Physically, this solution corresponds to a specific linear combination of modes that interfere in such a way that there is no transmission to the normal region. This requirement fixes the coefficients $c_n$ in (42). We will see that

this solution is already sufficient to determine the whole $S$ matrix. We find up to an overall constant factor,

$$\phi_2(z \to -\infty) \simeq \frac{e^{-\frac{2}{3}\sqrt{-2\alpha z^3}}(-z)^{-\frac{i\omega}{2\alpha}}}{\sqrt{-2\alpha z}}, \tag{43}$$

for $c_0 = 0$ and

$$c_1 = \frac{(2\alpha/9)^{\frac{i\omega}{6\alpha}}}{\sqrt{3\pi}}\left(\frac{4}{3\alpha}\right)^{\frac{1}{3}}\Gamma(a_1)\cos\left[\frac{\pi}{3}\left(1+\frac{i\omega}{\alpha}\right)\right], \tag{44}$$

$$c_2 = \frac{(2\alpha/9)^{\frac{i\omega}{6\alpha}}}{\sqrt{3\pi}}\,2\,\Gamma(a_2)\cosh\frac{\pi\omega}{3\alpha}, \tag{45}$$

$$c_3 = \frac{(2\alpha/9)^{\frac{i\omega}{6\alpha}}}{\sqrt{3\pi}}\,(6\alpha)^{\frac{1}{3}}\Gamma(a_3)\cos\left[\frac{\pi}{3}\left(1-\frac{i\omega}{\alpha}\right)\right]. \tag{46}$$

With the integration constants fixed, we find at the other side of the horizon,

$$\phi_2(z \to +\infty) \simeq c_u\,\frac{z^{\frac{i\omega}{\alpha}}}{\sqrt{\alpha z}} + c_{w_1}\frac{e^{+i\frac{2}{3}\sqrt{2\alpha z^3}}z^{-\frac{i\omega}{2\alpha}}}{\sqrt{-2\alpha z}} + c_{w_2}\frac{e^{-i\frac{2}{3}\sqrt{2\alpha z^3}}z^{-\frac{i\omega}{2\alpha}}}{\sqrt{-2\alpha z}}, \tag{47}$$

with

$$c_u = \frac{\sqrt{2\pi}\,(2\alpha)^{\frac{i\omega}{2\alpha}}}{\Gamma\left(\frac{1}{2}+\frac{i\omega}{\alpha}\right)}, \qquad c_{w_{1,2}} = \pm e^{\pm\frac{\pi\omega}{2\alpha}}, \tag{48}$$

which correspond to the matching coefficients of the WKB modes. Indeed, in the linear regime, away from the turning point, the WKB modes are given by

$$\frac{e^{i\int^z dz' k_u(z')}}{\sqrt{v_u}}\,\varphi_u \simeq \frac{z^{\frac{i\omega}{\alpha}}}{\sqrt{\alpha z}}\begin{pmatrix}-\frac{\omega}{2\alpha z}\\1\end{pmatrix}, \tag{49}$$

$$\frac{e^{i\int^z dz' k_{w_{1,2}}(z')}}{\sqrt{v_{w_{1,2}}}}\,\varphi_{w_{1,2}} \simeq \frac{e^{\pm i\frac{2}{3}\sqrt{2\alpha z^3}}z^{-\frac{i\omega}{2\alpha}}}{\sqrt{-2\alpha z}}\begin{pmatrix}\mp\sqrt{\frac{\alpha z}{2}}\\1\end{pmatrix}, \tag{50}$$

where Eq. (50) holds for $\ln z \gg \alpha z/2$. Note that we used the low-energy forms of the wavevectors, which hold away from the turning point $z \gg z_c$.

Hence, we find that the WKB solution and the approximate solution in the vicinity of the horizon both hold in a region where $|z-z_c| \gg 1$ and $|z| \ll 1/\alpha$ are satisfied simultaneously. We demonstrate this explicitly in Fig. 3 where we show the exact solution, obtained from numerically solving the stationary wave equation, together with the approximate solution near the horizon and the WKB solution. Here, the total WKB solution is given by

$$\phi_s^{(\text{WKB})}(z) = \sum_\mu c_\mu \phi_{\mu s}(z_0)\frac{\sqrt{v_\mu(z_0)}}{\varphi_{\mu s}(z_0)}\frac{e^{i\int_{z_0}^z dz' k_\mu(z')}}{\sqrt{v_\mu(z)}}\,\varphi_{\mu s}(z), \tag{51}$$

where $s = 1, 2$ corresponds to the spinor components and the sum runs over $\mu = \{w_2\}$ with $c_{w_2} = 1$ outside the horizon and $\mu = \{u, w_1, w_2\}$ with the $c_\mu$ given in Eq. (48) inside the horizon. Here, $\phi_{\mu s}(z_0)$ are the WKB modes in the linear regime and the fitting parameter $z_0 < 0$ ($z_0 > 0$) outside (inside) the horizon. For simplicity, we take the same $z_0$ for all modes in a given region. We then optimize its value by hand in a region where both solutions should approximately hold. In the figure, we see that the approximations match reasonably well to the exact solution. While the WKB solution breaks down near the classical turning point and the horizon at the origin, the approximate solution near the horizon fits perfectly in the type-I region, but starts to fail in the type-II region away from the horizon. In general,

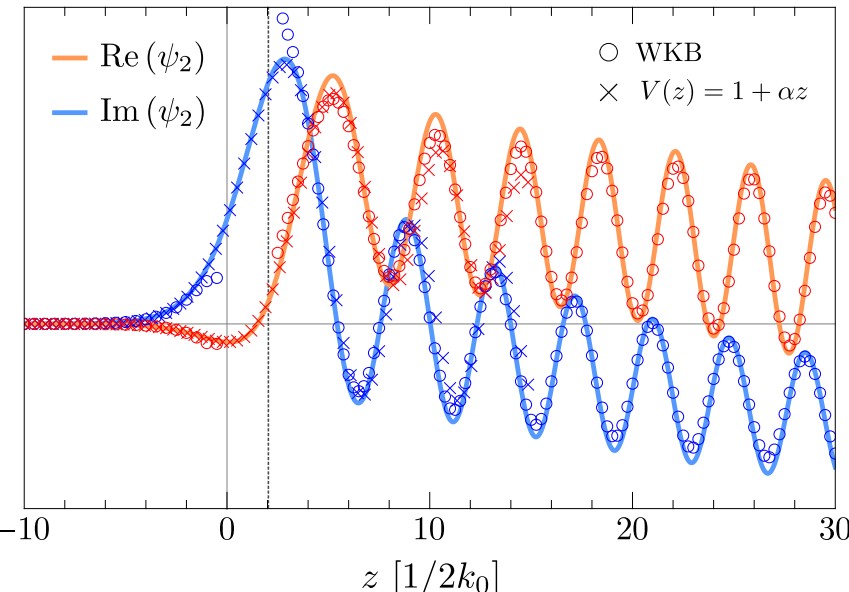

Figure 3: Real and imaginary part of the second spinor component $\psi_2$ for the case of zero transmission outside of the horizon and the tilt profile $V(z) = 1 + \tanh(\alpha z)$ with $\alpha = 0.1$ and $\omega = 0.05$. Solid curves give the exact solution, while the circles and crosses give the WKB solutions and the approximate solution near the linear horizon, respectively. The vertical dotted line marks the classical turning point.

matching becomes worse with increasing $|\omega|$ and $\alpha$ as expected from our assumptions. Inside the horizon, the wave function features an envelope from the $u$ mode which has a relatively long wavelength, while the oscillations inside the envelope are due to the short-wavelength $w$ modes (Fig. 2).

The solution that we obtained in the previous section is purely decaying in the type-I region such that $b_{Lu} = 0$ and the scattering coefficients are related by

$$
\begin{pmatrix} 0 \\ b_{Ru} \\ b_{Rv} \end{pmatrix} = \begin{pmatrix} t_{uw_1} & t_{uw_2} & 0 \\ r_{uw_1} & r_{uw_2} & 0 \\ 0 & 0 & 1 \end{pmatrix} \begin{pmatrix} a_{Rw_1} \\ a_{Rw_2} \\ a_{Lv} \end{pmatrix} ,
\tag{52}
$$

where we already took into account that the $v$ modes are decoupled from the $u$ and $w$ modes for a slowly-varying tilt profile. From the first equation of (52), we obtain

$$
\frac{t_{uw_2}}{t_{uw_1}} = -\frac{a_{Rw_1}}{a_{Rw_2}} = e^{\frac{\pi\omega}{\alpha}} ,
\tag{53}
$$

where we used Eq. (48). Together with the unitarity of the $S$ matrix, we find

$$
S_{\text{counter}} = \frac{(2\alpha)^{\frac{i\omega}{2\alpha}} e^{-\frac{\pi\omega}{2\alpha}} \Gamma\left(\frac{1}{2} - \frac{i\omega}{\alpha}\right)}{\sqrt{2\pi}} \begin{pmatrix} 1 & e^{\frac{\pi\omega}{\alpha}} \\ e^{\frac{\pi\omega}{\alpha}} & -1 \end{pmatrix} ,
\tag{54}
$$

with scattering probabilities

$$
T_{u \leftarrow w_{1,2}} = R_{u \leftarrow w_{2,1}} = \frac{1}{1 + e^{\pm\frac{2\pi\omega}{\alpha}}} ,
\tag{55}
$$

which have the form of a Fermi-Dirac distribution with an effective Hawking temperature $T_H = \hbar v \alpha / 2\pi k_B$. Corrections to this low-energy result yield an energy-dependent Hawking

temperature $T_H(\omega) = T_H(-\omega)$. These analytical results agree well with numerical lattice calculations (see App. B).

Note that the transmission saturates for $|\omega| \approx \alpha$ such that tunneling through the horizon occurs only for $|\omega| < \alpha$. We can understand this by noting that the $S$ matrix has simple poles at

$$\omega = -i\alpha\left(n + \frac{1}{2}\right), \qquad (n = 0, 1, 2, \ldots), \tag{56}$$

which correspond to quasi-bound states [28]. Classically, an incoming $w_1$ or $w_2$ particle is completely reflected at the turning point and the classical contribution to the transmission is a step function $\Theta(\mp\omega)$. However, quantum-mechanically the $w$ particles can tunnel through the horizon via a transient state with lifetime $1/\alpha$.

Before we proceed with the implications of these results in a two-terminal transport setup, we first consider the opposite limit where the tilt profile is sharp relative to the Fermi wavelength. We will demonstrate that in this case the co- and counterpropagating modes are coupled by the horizon. Nevertheless, one can still define a Hawking temperature at low energies, even though the transmission is not a thermal distribution in this case.

## 3.2 Sharp tilt profile

When the tilt profile is sharp on the scale of the Fermi wavelength (i.e., the limit $\alpha \gg 1$), an incoming wave packet cannot resolve the precise details of the interface and we can model the tilt profile with a step function

$$V(z) = V_L\Theta(-z) + V_R\Theta(z), \tag{57}$$

where $\Theta$ is the Heaviside step function. Assuming the wave function is continuous at $z = 0$, we integrate the wave equation $\mathcal{H}_0\phi = \omega\phi$ over an infinitesimal region of length $2\epsilon$ centered at the origin. This gives

$$[-i\partial_z\phi]_{-\epsilon}^{+\epsilon} + \frac{V_R - V_L}{2}\sigma_x\,\phi(0) = 0, \tag{58}$$

where we used $\partial V/\partial z = (V_R - V_L)\delta(z)$. These boundary conditions are physically sound since they keep the current density continuous

$$j_R = \left[\phi^\dagger(V_R\sigma_0 + \sigma_z)\phi + 2\,\text{Im}\left(\phi^\dagger\sigma_x\partial_z\phi\right)\right]_{z=+\epsilon} \tag{59}$$

$$= \left[\phi^\dagger(V_R\sigma_0 + \sigma_z)\phi + 2\,\text{Im}\left(\phi^\dagger\sigma_x\partial_z\phi\right)\right]_{z=-\epsilon} - (V_R - V_L)|\phi(0)|^2 \tag{60}$$

$$= \left[\phi^\dagger(V_L\sigma_0 + \sigma_z)\phi + 2\,\text{Im}\left(\phi^\dagger\sigma_x\partial_z\phi\right)\right]_{z=-\epsilon} = j_L. \tag{61}$$

We now consider a black hole horizon, i.e., we take $0 \leq V_L < 1$ and $V_R > 1$. Since the tilt is constant in each region, the wave function is given by a superposition of plane waves. In the type-I region ($z < 0$), we obtain

$$\Phi_L(z) = \frac{a_{Lv}}{\sqrt{v_{Lv}}}\varphi_{Lv}e^{ik_{Lv}z} + \frac{b_{Lu}}{\sqrt{-v_{Lu}}}\varphi_{Lu}e^{ik_{Lu}z} + c_L\,\varphi_{Lw}e^{ik_{Lw}z}, \tag{62}$$

with $\text{Im}\,k_{Lw} < 0$ and where $a$, $b$, and $c$ are coefficients of incoming, outgoing, and evanescent modes. Here, we also normalized the scattering states such that each mode contributes unit current. Up to first order in $\omega$, the group velocities are given by

$$v_v \simeq V + 1, \tag{63}$$

$$v_u \simeq V - 1, \tag{64}$$

$$v_{w_{1,2}} \simeq \frac{1 - V^2}{V} \pm \frac{2 + V^{-2}}{\sqrt{V^2 - 1}}\,\omega. \tag{65}$$

In the type-II region ($z > 0$), we find for $|\omega| < \omega_c$,

$$\Phi_R(z) = \frac{b_{Rv}}{\sqrt{v_{Rv}}}\,\varphi_{Rv}e^{ik_{Rv}z} + \frac{b_{Ru}}{\sqrt{v_{Ru}}}\,\varphi_{Ru}e^{ik_{Ru}z} + \frac{a_{Rw_1}}{\sqrt{-v_{Rw_1}}}\,\varphi_{Rw_1}e^{ik_{Rw_1}z} + \frac{a_{Rw_2}}{\sqrt{-v_{Rw_2}}}\,\varphi_{Rw_2}e^{ik_{Rw_2}z},$$

(66)

such that in this case the $S$ matrix can be written as

$$\begin{pmatrix} b_{Lu} \\ b_{Ru} \\ b_{Rv} \end{pmatrix} = \begin{pmatrix} t_{uw_1} & t_{uw_2} & r_{uv} \\ r_{uw_1} & r_{uw_2} & t_{uv} \\ r_{vw_1} & r_{vw_2} & t_{vv} \end{pmatrix} \begin{pmatrix} a_{Rw_1} \\ a_{Rw_2} \\ a_{Lv} \end{pmatrix}.$$

(67)

On the other hand, for $|\omega| > \omega_c$, the solution in the overtilted region also consists of two scattering states and one evanescent mode. We do not discuss this regime here, as we are mostly interested in the low-energy physics.

The $S$ matrix is then determined as usual. Namely, by setting all but one of the incoming coefficients zero, and calculating the outgoing coefficients with the boundary conditions at the origin. Note that one must include the evanescent modes to obtain a unique solution. In principle, one can obtain closed form expressions for the scattering coefficients but this is cumbersome as the wavevectors are given by the roots of a fourth-order polynomial. When the incoming mode comes from the effective black hole region behind the horizon, we find up to first order in $\omega$,

$$T_{u \leftarrow w_{1,2}} \simeq \frac{8(V_L + V_R)}{(V_L + V_R + 2)^2}\left(\frac{1}{2} \mp \frac{\omega}{4k_B T_u}\right),$$

(68)

$$R_{v \leftarrow w_{1,2}} \simeq \frac{(V_L - V_R)^2}{(V_L + V_R + 2)^2}\left(\frac{1}{2} \pm \frac{\omega}{4k_B T_v}\right),$$

(69)

and $R_{u \leftarrow w_{1,2}} = 1 - T_{u \leftarrow w_{1,2}} - R_{v \leftarrow w_{1,2}}$ with

$$T_u = \frac{1}{2k_B}\frac{\left(1 - V_L^2\right)(V_R - V_L)\left(V_R^2 - 1\right)^{3/2}}{2 + V_R^4 + 3V_L^3 V_R + V_L^2\left(1 - 4V_R^2\right) + V_L V_R\left(2V_R^2 - 5\right)},$$

(70)

$$T_v = \frac{1}{2k_B}\frac{(V_R - V_L)\left(V_R^2 - 1\right)^{3/2}}{3V_R\left(V_R^2 - 2\right) + V_L\left(V_R^2 + 2\right)},$$

(71)

which are the effective Hawking temperatures of intrabranch and interbranch processes, respectively, for the sharp tilt profile ($\alpha \gg 1$) [3]. This interpretation rests on our results for the slowly-varying limit ($\alpha \ll 1$) and demonstrates that some aspect of analog Hawking radiation survives for the sharp tilt profile at low energies.

These low-energy expressions are compared to the exact results in Fig. 4. Note that the sharp horizon couples co- and counterpropagating modes, e.g., through scattering processes such as $w_1(w_2) \to v$ although these processes are generally suppressed. As we discussed in the previous section, such processes become negligible in the slowly-varying tilt profile. Moreover, unlike in the slowly-varying limit, the transmission functions now depend explicitly on the asymptotic values of the tilt profile. Similar expressions for the transmission functions can be obtained when the incoming mode is incident on the horizon from the normal (type-I) region, although the first-order term vanishes in this case. The complete $S$ matrix for the sharp horizon at low energies is given in App. C.

## 4  Hawking effect out of equilibrium

In equilibrium, there is no net Hawking current as the particle number is conserved in our system, unlike for an actual black hole which provides an energy source for particle creation

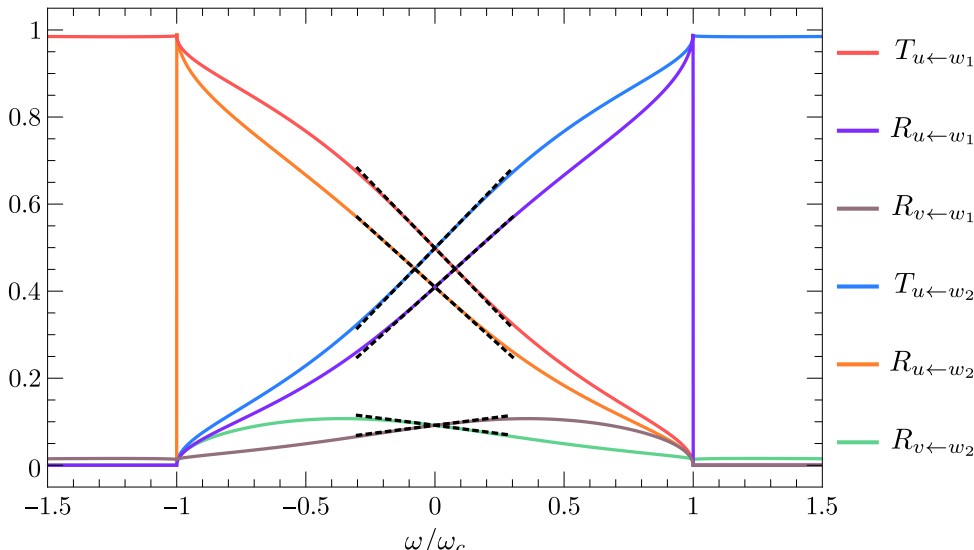

Figure 4: Transmission and reflection probabilities for an incoming mode incident on the sharp horizon from the black hole (type II) region for $(V_L, V_R) = (0.2, 2)$. Dashed lines correspond to the first-order analytical results given in Eqs. (68) and (69).

[10,29]. Furthermore, as long as the type-II region is in local equilibrium, the total current out of the black hole region, summing contributions from $w_1$ and $w_2$ modes, is always ballistic (in the absence of disorder), and the two-terminal conductance is simply a measure of the density of states. In order to obtain a net Hawking current, we require a non-equilibrium occupation in the type-II region. For example, if the type-II phase is induced in a quenched way [10], $V(t) = V_0 \Theta(t)$ with $V_0 > 1$, one obtains a transient state with excited $w_1$ modes above the Fermi level and empty $w_2$ states below the Fermi level, giving rise to a net transmission of $w$ modes.

Here, we propose two alternative ways of achieving a non-equilibrium situation by taking advantage of the spin structure of the $w$ modes. In particular, we demonstrate that one can favor populating $w_1$ over $w_2$ by exciting a photocurrent with circularly-polarized light or by injecting a spin-polarized current from a magnetic lead. In both cases, the occupation of one of the $w$ modes is favored, which will then tunnel through the horizon from the type-II region, giving rise to a net Hawking current in the type-I region, assuming relaxation due to, e.g., disorder, within the type-II region occurs over a sufficiently long time scale such that the favored $w$ mode can be transmitted across the horizon. In fact, transport lifetimes up to $\tau \approx 45$ ps have been measured in the type-I Weyl semimetal TaAs [30], giving rise to a rather long mean free path of 5.2 $\mu$m.

## 4.1 Irradiation by circularly-polarized light

We can favor the occupation of one of the $w$ modes by irradiating the type-II region. This gives rise to optical transitions inside the overtilted region from $v$ modes to $w_2$ modes for $V/v_F > 1$ (Fig. 2). In this case, there are no states available for optical transitions to the $w_1$ modes (we approximate optical transitions to be effectively momentum-conserving). This continues to hold in the presence of more bands as long as the energy separation to the lower bands is of a different energy range than the photon energy of interest.

For concreteness, we consider circularly-polarized light of frequency $\Omega$ traveling in the $z$ direction, as illustrated in Fig. 1(a). Here, we assume that the light hits the type-II Weyl semimetal sufficiently far away from the horizon where the tilt $V/v_F > 1$ is essentially constant.

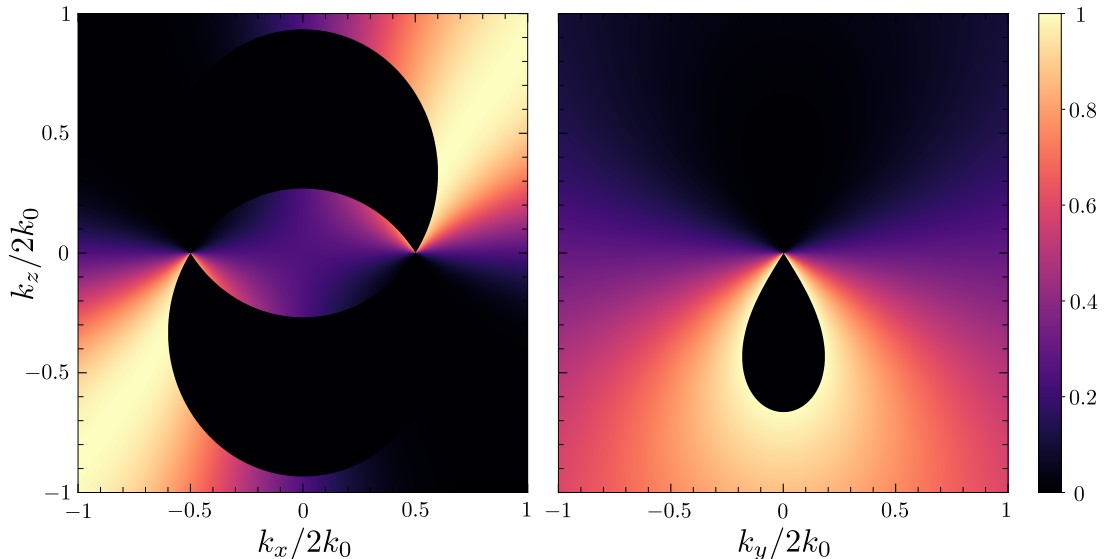

Figure 5: Out-of-equilibrium occupation $n_{k+}$ of the conduction band, integrated over the photon energies, in units of $4\pi\tau(e v_F A_0)^2/\hbar$ for $k_y = 0$ (left) and at $k_x = -k_0$ (right). The black region for negative (positive) $k_z$ denotes an electron (hole) pocket where optical transitions are forbidden for $E_F = 0$. The occupied states with negative $k_z$ away from the electron pockets have negative group velocities and are transmitted across the horizon.

The light-matter interaction is treated classically with the vector potential

$$A(t) = A_+ e^{-i\Omega t} + A_- e^{i\Omega t}, \tag{72}$$

where $A_\pm = A_0 \left(e_x \pm i e_y\right)/\sqrt{2}$. Note that we discard Zeeman coupling which is suppressed by a factor $10^{-3}$ compared to orbital coupling in generic Weyl semimetals [31]. Letting $k \to k + eA/\hbar$ in (12) gives the light-matter interaction $V_\pm = e J \cdot A_\pm$ to first order in the electron charge $-e$. These terms describe transitions between occupied and unoccupied modes via absorption ($V_+$) and emission ($V_-$). Here, we defined the current operator $J$ with components

$$J_x = \frac{v_F}{k_0} k_x \sigma_x, \quad J_y = \frac{v_F}{k_0} k_y \sigma_x + v_F \sigma_y, \quad J_z = \frac{v_F}{k_0} k_z \sigma_x + v_F \sigma_z + V \sigma_0. \tag{73}$$

The rate of change of the distribution function $n_{k\lambda}$ of the $\lambda = \pm$ band is obtained from the Boltzmann equation in the relaxation-time approximation [32],

$$\frac{\partial n_{k\lambda}}{\partial t} + \dot{k} \cdot \frac{\partial n_{k\lambda}}{\partial k} + \dot{r} \cdot \frac{\partial n_{k\lambda}}{\partial r} = \Gamma_k(\Omega)(n_{k-\lambda} - n_{k\lambda}) - \frac{n_{k\lambda} - n_{k\lambda}^0}{\tau}, \tag{74}$$

where the relaxation time $\tau$ takes into account intraband impurity and phonon scattering [33] and $\Gamma_k(\Omega)$ gives the rates for vertical transitions, calculated with Fermi's golden rule,

$$\Gamma_k(\Omega) = \frac{2\pi}{\hbar} |\langle \varphi_{k+} | V_+ | \varphi_{k-} \rangle|^2 \, \delta \left( E_+(k) - E_-(k) - \hbar\Omega \right), \tag{75}$$

where the dispersion relation is given by

$$E_\pm(k) = 2\hbar v_F k_0 \left[ \frac{V}{v_F} \frac{k_z}{2k_0} \pm d(k) \right], \tag{76}$$

with $d = \sqrt{[(k/2k_0)^2 - 1/4]^2 + (k_y/2k_0)^2 + (k_z/2k_0)^2}$ and $k = |\boldsymbol{k}|$. If we Fourier transform Eq. (74) and consider the DC limit, as well as the long wavelength limit ($\partial_r n_{\boldsymbol{k}} \approx 0$), the non-equilibrium occupation is given in lowest order of $A_0$ by

$$n_{\boldsymbol{k}\lambda} \simeq n_{\boldsymbol{k}\lambda}^0 + \tau \Gamma_{\boldsymbol{k}}(\Omega)\left(n_{\boldsymbol{k}-\lambda}^0 - n_{\boldsymbol{k}\lambda}^0\right). \tag{77}$$

The transition matrix elements writes:

$$|\langle \varphi_{\boldsymbol{k}+}| V_+ |\varphi_{\boldsymbol{k}-}\rangle|^2 = 2(ev_F A_0)^2 F(\boldsymbol{k}), \qquad F(\boldsymbol{k}) = \frac{\left[d + k_x k_z/(2k_0^2)\right]^2}{4d^2}, \tag{78}$$

where $0 \le F \le 1$ and which at the Weyl nodes $\boldsymbol{k}_\perp^{(\pm)} = (\pm k_0, 0)$ reduces to

$$F(k_z) = \frac{\left[c(k_z) \pm \mathrm{sgn}(k_z)\right]^2}{4c(k_z)^2}, \tag{79}$$

where $c(k_z) = \sqrt{1 + (k_z/2k_0)^2}$. Since there are no states below the $w_1$ modes, optical transitions are absent and the occupation above the Fermi energy vanishes at zero temperature, i.e., $\Delta n_{w_1} = 0$. On the other hand, for $w_2$ modes, at zero temperature, the difference in occupation between the two bands vanishes inside the electron pocket which corresponds to $k_{w_2}(E_F) \le k_z \le 0$ where $E_F$ is the Fermi energy and $k_{w_2}$ is given by Eq. (22). The occupation of $w_2$ modes for $\boldsymbol{k} = (\boldsymbol{k}_\perp^{(\pm)}, k_z)$ is thus given by

$$\Delta n_{w_2} = \frac{4\pi\tau(ev_F A_0)^2}{\hbar} F(k_z) \Theta\left[k_{w_2}(E_F) - k_z\right] \delta\left(\hbar v_F |k_z| c(k_z) - \hbar\Omega\right), \tag{80}$$

such that a net occupation imbalance is generated. Away from $\boldsymbol{k}_\perp = \boldsymbol{k}_\perp^{(\pm)}$, we plot in Fig. 5 the non-equilibrium occupation, integrated over the energy, in units of $4\pi\tau(ev_F A_0)^2/\hbar$. As we see in the right figure, for the Weyl node at $k_x = -k_0$, the occupation probability of the $w_2$ modes with negative group velocity along the $z$ axis *increases*. Hence, provided the energy of the photons matches the transition energy, only these $w_2$ modes are transmitted across the horizon. On the other hand, since $n_{(-k_x, k_y, k_z)+} = n_{(k_x, k_y, -k_z)+}$, the occupation probability for the second Weyl node at $k_x = +k_0$ will *decrease* for negative $k_z$, as is already apparent from the left hand side of Fig. 5. As such, only one of the overtilted Weyl cones contributes to the fermionic Hawking effect out of equilibrium. The non-equilibrium occupation of the $u$ modes above the Fermi energy in the undertilted region is then given by

$$n_{L,u} = T_{u \leftarrow w_1} n_{R,w_1} + T_{u \leftarrow w_2} n_{R,w_2} = T_{u \leftarrow w_2} \Delta n_{w_2}, \tag{81}$$

where the first equality holds only for a slowly-varying tilt profile.

## 4.2 Coupling to magnetic leads

An out-of-equilibrium distribution can also be induced by coupling the overtilted region to magnetic leads. Here, we assume that the Pauli matrices in Hamiltonian (12) effectively correspond to the physical spin degrees of freedom of the electrons. In fact, for Weyl semimetals with broken time-reversal symmetry, but with inversion and cubic symmetries, this correspondence is exact [34]. In general, they also contain other degrees of freedom like orbital or lattice degrees of freedom. Moreover, models for which $\boldsymbol{\sigma}$ corresponds to the real spin have been shown to simulate spin textures of Weyl semimetals observed in experiments [35, 36].

For simplicity, we consider the 1D effective model for $\boldsymbol{k}_\perp = (\pm k_0, 0)$. In this case, the electrons will be effectively polarized in the $xz$ plane with different orientations for the $w_1$

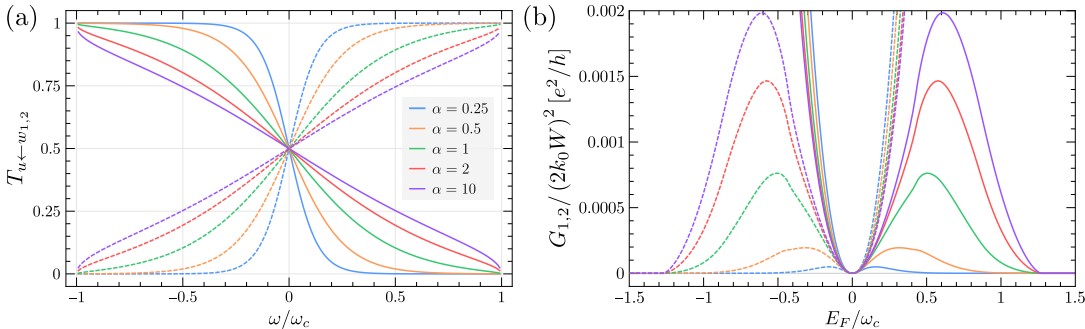

Figure 6: (a) $T_{u \leftarrow w_1}$ (solid) and $T_{u \leftarrow w_2}$ (dashed) for $\boldsymbol{k}_\perp = (\pm k_0, 0)$ calculated with the lattice model for the tilt profile $V(z) = v_F [1 + \tanh(\alpha z)]$ where $\omega_c \approx 0.6 \times 2\hbar v_F k_0$ in the type-II region. (b) Zero-bias differential conductance at zero temperature $G_1$ (solid) and $G_2$ (dashed) for the same parameters as in (a).

and $w_2$ modes. This is already apparent in Eq. (25) where the modes have different spin projections along $k_z$. The magnetic leads can be modeled, for instance, by a 1D chiral fermion with spin-dependent group velocities, giving rise to constant but different density of states for the two spin bands.

We further assume that the spin-up lead electrons have the same polarization as the $w_1$ electrons at $\omega = 0$. In this case, the spinors $\psi_{\sigma = \pm}$ of the magnetic lead can be written as $\psi_+ = \varphi_{w_1}$ and $\psi_- = i\sigma_y \varphi_{w_1}$, with $\varphi_{w_1}$ given in Eq. (25). A bias voltage $U_{DC}$ is applied between the magnetic lead and the overtilted region by setting the chemical potential to $\mu_R = eU_{DC}$ inside the lead for both spin species, and to $\mu_L = 0$ inside the type-II Weyl semimetal. The overlap between the lead spinors and the Weyl semimetal spinor will then give rise to an effective tunneling Hamiltonian [37],

$$H_T = \sum_{k,k'} \sum_{\sigma \lambda} t_{k\lambda}^\sigma \hat{c}_{k\lambda}^\dagger \hat{f}_{k'\sigma} + h.c., \tag{82}$$

where $\hat{f}_{k'\sigma}$ destroys a spin-$\sigma$ electron with momentum $k'$ inside the lead, $\hat{c}_{k\lambda}^\dagger$ creates a Weyl mode inside the overtilted region, and $t_{k\lambda}^\sigma = t_0 \varphi_{k\lambda}^\dagger \psi_\sigma$ are the tunneling matrix elements with tunneling strength $t_0$. By construction, the spin-up electrons couple much stronger to the $w_1$ modes than to the $w_2$ modes. Hence, tunneling between the magnetic lead and the overtilted Weyl semimetal will give rise to a non-equilibrium occupation. Following the previous section, the occupation can be modeled with the Boltzmann equation in the relaxation-time approximation, which essentially leads to Eq. (77). Assuming the density of states of the spin-up lead electrons $g_\uparrow \gg g_\downarrow$, one finds at zero temperature,

$$n_{k\lambda} = n_{k\lambda}^0 + \tau \frac{2\pi}{\hbar} g_\uparrow |t_{k\lambda}^\uparrow|^2 \Big[ \Theta(eU_{DC} - E_\lambda(k)) - \Theta(-E_\lambda(k)) \Big], \tag{83}$$

where the dispersion is given by Eq. (76). In particular, for our choice of polarization of the magnetic lead, we have $|t_{w_2}^\uparrow|^2 / |t_{w_1}^\uparrow|^2 \approx v_F^2 / V^2$ for states close to $\omega = 0$. Hence, the occupation of the $w_2$ modes is suppressed by the tilt $V$. This again gives rise to a population imbalance between the $w_{1,2}$ modes, such that the Hawking signature can in principle be observed.

## 4.3 Differential conductance

In Section 3, we considered the case of normal incidence with $\boldsymbol{k}_\perp = (\pm k_0, 0)$ whose contribution dominates at low energies. However, in a transport experiment all transverse channels contribute to the current. Hence, it is not clear what remains of the Hawking effect even if one

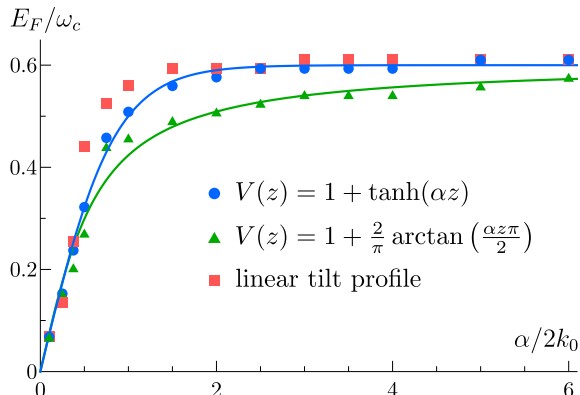

Figure 7: Position of the local maximum in the zero-bias differential conductance for the $w_1$ mode, as a function of the Fermi energy $E_F$ and the slope of the tilt profile $\alpha$, for different tilt profiles (in units of $v_F$) as detailed in App. B. Here, the solid blue and green curves are fits of the data to $a \tanh b\alpha$ and $\frac{2a}{\pi} \arctan \frac{\pi b \alpha}{2}$, respectively, with $a \approx 0.6$ and $b \approx 1.2$.

can excite an $I_{w_i}$ ($i = 1, 2$) current by means of a non-equilibrium occupation of $w$ modes, as described above. We therefore calculate the contribution of the $w_i$ mode to the two-terminal zero-bias differential conductance at zero temperature,

$$G_i(E_F) \equiv \left. \frac{dI_{w_i}}{dV} \right|_{V=0,T=0} = \frac{e^2}{h} \sum_{\boldsymbol{k}_\perp} T_{u \leftarrow w_i}(\omega = E_F, \boldsymbol{k}_\perp) \tag{84}$$

$$= \frac{e^2}{h} \frac{W^2}{2\pi^2} \int_0^\infty dk_x \int_{-\infty}^\infty dk_y \, T_{u \leftarrow w_i}(\omega = E_F, \boldsymbol{k}_\perp), \tag{85}$$

where we take a sample with transverse dimensions $W \times W$ and we used $\mathcal{H}(k_x) = \mathcal{H}(-k_x)$ in the second line. Here, we reverted to dimensionful units for clarity. The transmission functions for general $\boldsymbol{k}_\perp$ are calculated with the KWANT Python package (see App. B).

The transmission of the Hawking channel ($k_x = \pm k_0$ and $k_y = 0$) obtained with the lattice model is shown in Fig. 6(a). In the slowly-varying limit ($\alpha \ll 1$), the transmission matches perfectly our analytical results given in Eq. (55). On the other hand, for $\alpha \gg 1$, we also find agreement with our results for the sharp horizon. The conductance for the $w_1$ and $w_2$ current is shown in Fig. 6(b) as a function of the Fermi energy. Here, the quadratic behavior near the Weyl node is attributed to the low-energy density of states in the type-I region, given by $g(\omega) = v_F \omega^2 / [2\pi^2 \hbar^3 (v_F^2 - V_L^2)^2]$. Interestingly, we observe that the conductance $G_1$ ($G_2$) features a local maximum at positive (negative) energies, due to tunneling of $w_1$ ($w_2$) modes through the effective horizon. The peak position in the $(\alpha, E_F)$ plane is shown in Fig. 7 for three different tilt profiles. In all cases, the peak position increases linearly for small $\alpha$ at first, with the same slope $E_F / \omega_c \approx 0.7 \alpha / 2k_0$. Furthermore, in the limit of a sharp horizon ($\alpha \gg 1$), the same constant is attained, given by $E_F / \omega_c \approx 0.6$ and whose precise value depends only on the asymptotic values of the tilt profile. Surprisingly, we find that the position of the maxima as a function of $\alpha$ are fitted reasonably well to functions that are similar to the tilt profile, as illustrated in Fig. 7, except in the case with the purely linear tilt profile.

We thus conclude that it is in principle possible to induce a stationary non-equilibrium occupation of $w$ modes, either by irradiation of light in the type-II region, or by coupling the type-II region to a magnetic lead. In both cases, we assume this takes place in the asymptotic tilt region far away from the horizon. For sufficiently long mean-free paths this then gives rise to a net $w_1$ or $w_2$ current. The corresponding two-terminal differential conductance features

a peak due to tunneling of $w_1$ or $w_2$ modes across the horizon whose height and position depends in general on all the details of the tilt profile. However, for a slowly-varying (sharp) tilt profile relative to the Fermi wavelength, the properties of the peak depend only on the slope $\alpha$ (the asymptotic values of the tilt).

# 5 Conclusion

In this work, we investigated mesoscopic transport across effective event horizons at the interface of a type-I and type-II Weyl semimetal. To this end, we used a minimal model that captures all salient features of a Weyl semimetal, and studied type-I/type-II interfaces with different Weyl node tilt profiles. We solved the scattering problem analytically at normal incidence in the low-energy limit for two cases: a fast and slow varying tilt profile. More precisely, these two cases are distinguished by the length scale of the tilt profile relative to the Fermi wavelength.

For a slowly-varying tilt profile, we employed the WKB formalism together with an approximate solution near a linear horizon. We find that co- and counterpropagating modes are decoupled in this limit and we calculated the $S$ matrix, which depends only on the energy and the slope of the tilt profile at the horizon. The irrelevance of further microscopic details is reminiscent of a "no-hair theorem" for real black holes. Moreover, the transmission functions of counterpropagating modes are given by a thermal distribution with effective Hawking temperature inversely proportional to the slope. Adding the different contributions of the counter-propagating modes to transport in a Landauer-Büttiker picture, however, masks all analogs of Hawking effects.

For the sharp horizon, we solved the scattering problem by first deriving appropriate boundary conditions for the wavefunction at the horizon. In this case, the $S$ matrix explicitly depends on the asymptotic values of the tilt profile. Hence, the black hole analogy breaks down in this case, even though one can still define a temperature scale.

To circumvent the ballistic nature of transport in the slowly-varying limit, we considered means to drive a non-equilibrium occupation of $w$ modes, i.e., those modes that tunnel through the horizon from inside the type-II region. We showed that one can favor populating one of the $w$ modes over the other by irradiating the type-II region with circularly polarized light, or by coupling it to a magnetic lead. Given this non-equilibrium occupation, we then calculated the differential conductance for a single $w$ mode which displays a peak as a function of the Fermi energy. The peak position becomes universal in the limit of very slowly varying tilts: it only depends on the slope of the tilt profile at the horizon. In the opposite limit of rapidly changing tilts, the peak position saturates to a value that only depends on the asymptotic values of the tilt. However, in the intermediate regime, the details of the whole tilt profile become important.

The transport experiment we propose can serve as a proof of principle for analog event horizons in fermionic systems. Additionally, to better distinguish different tilting regimes, one may apply a magnetic field along the tilting direction, to freeze out all transverse degrees of freedom save the Hawking channel. In conclusion, we have proposed how to detect signatures of analog Hawking radiation in a type-I and type-II Weyl semimetal heterostructure by driving the system out of equilibrium. The proposed setup has potential further applications in terms of electron lensing, which can be readily understood through the effective spacetime and associated gravitational lensing analogies.

# Acknowledgements

The authors are indebted to Christian Schmidt and Andreas Haller for insightful discussions.

**Funding information** CDB, SG, and TLS acknowledge support by the National Research Fund Luxembourg under the grants ATTRACT 7556175 and PRIDE/15/10935404. TM acknowledges financial support by the Deutsche Forschungsgemeinschaft via the Emmy Noether Programme ME4844/1-1 (project id 327807255), the Collaborative Research Center SFB 1143 (project id 247310070), and the Cluster of Excellence on Complexity and Topology in Quantum Matter ct.qmat (EXC 2147, project id 390858490).

*Note:* A related publication in Ref. [38] provides a numerical analysis of wavepacket dynamics in microscopic lattice models of Weyl black and white hole analogs with a focus on lattice effects, and a discussion of realizations in metamaterials.

# Appendix

## A Covariant form of the Weyl equation

In this appendix, we explicitly demonstrate how the effective Weyl equation for a tilt profile $V = V(z)$ describing low-energy excitations of a tilted Weyl semimetal can be cast into a manifestly covariant form using the tetrad formalism [10,20].

Unlike vectors and tensors, the Lorentz transformation rule for spinors in flat spacetime does not generalize to curved spacetime, because the group of real invertible matrices $GL(4, \mathbb{R})$ has no spinor representations [21,39]. However, we can introduce local Lorentz frames at each point in spacetime and define spinors with respect to these frames. To this end, we choose an orthonormal basis of the tangent space $\hat{e}_a(p)$ ($a = 0, 1, 2, 3$) at each point $p$ of spacetime, i.e.,

$$g(\hat{e}_a, \hat{e}_b) = \eta_{ab}, \tag{86}$$

where the vector fields $\hat{e}_a(p)$ are called frame fields or tetrads and $\eta_{ab}$ is the Minkowski metric of flat spacetime. Hence, we can think of tetrads as arising from a set of coordinate transformations $x^\mu \to \xi_p^a$, one for each point $p$, such that $g_{ab}(p) = \eta_{ab}$ or

$$\eta_{ab} = \frac{\partial x^\mu}{\partial \xi_p^a} \frac{\partial x^\nu}{\partial \xi_p^b} g_{\mu\nu}\bigg|_p = e^\mu_{\ a}(p) e^\nu_{\ b}(p) g_{\mu\nu}(p), \tag{87}$$

where $g_{\mu\nu}$ is the metric in the coordinate basis $\hat{e}_\mu = \partial_\mu$ and we use the Einstein summation convention. The $\xi_p^a$ coordinates are called local inertial coordinates and the tetrads $\hat{e}_a(p) = e^\mu_{\ a}(p) \hat{e}_\mu$ constitute a local Lorentz frame [40]. Note that tetrads are not unique, since a different choice of local inertial coordinates at each point $p$ of spacetime, $\xi_p^a \to \xi_p^{a'} = \Lambda^{a'}_{\ b}(p)\xi_p^b$ induces a local Lorentz transformation $\Lambda(p)$ such that $e^\mu_{\ a} \to e^\mu_{\ a'} = \Lambda^b_{\ a'}e^\mu_{\ b}$, where $\Lambda^b_{\ a'}$ is the inverse transformation, and which leaves Eq. (86) invariant.

To construct the covariant Dirac equation, we require a covariant derivative $\mathcal{D}_a\psi$ which is a local Lorentz vector that transforms as a spinor. It follows that [21]

$$\gamma^a \mathcal{D}_a \psi = 0, \tag{88}$$

where $\mathcal{D}_a = e^\mu_{\ a}\left(\partial_\mu + \Omega_\mu\right)$, $\gamma^a$ are the Dirac matrices with $\{\gamma^a, \gamma^b\} = 2\eta^{ab}$, and $\Omega_\mu$ is called the spin connection. The spin connection ensures covariance since $\partial_\mu \psi$ does not transform as a spinor under local Lorentz transformations. In general, we can write Eq. (88) as

$$i\partial_t \psi = \left(-i\alpha^i \partial_i - i\Upsilon\right)\psi \equiv \mathcal{H}\psi, \tag{89}$$

with

$$\alpha^i(p) = (e^0_{\ b}\gamma^b)^{-1}e^i_{\ a}\gamma^a, \qquad \Upsilon(x) = (e^0_{\ b}\gamma^b)^{-1}e^\mu_{\ a}\gamma^a \Omega_\mu, \tag{90}$$

where the spin connection ensures that the Hamiltonian $\mathcal{H}$ is Hermitian [14]. If we compare this to the Hamiltonian of the tilted Weyl semimetal (1), we identify

$$e^\mu_{\ a}(z) = \delta^\mu_a + V(z)\delta^\mu_3 \delta^0_a, \tag{91}$$

which correspond to the so-called acoustic metric [4]:

$$g_{\mu\nu} = \eta_{ab}e^a_{\ \mu}e^b_{\ \nu} = \begin{pmatrix} V^2 - 1 & 0 & 0 & -V \\ 0 & 1 & 0 & 0 \\ 0 & 0 & 1 & 0 \\ -V & 0 & 0 & 1 \end{pmatrix}, \tag{92}$$

where $e^a_{\ \mu}$ are the inverse tetrads. Equivalently, it can be obtained from the inverse metric $g^{\mu\nu} = \eta^{ab}e^\mu_{\ a}e^\nu_{\ b}$. Explicitly, the spin connection is given by

$$\Omega_\mu = \frac{1}{8}[\gamma^a, \gamma^b]g_{\kappa\lambda}e^\kappa_{\ a}(\partial_\mu e^\lambda_{\ b} + \Gamma^\lambda_{\mu\nu}e^\nu_{\ b}), \tag{93}$$

which can be derived from infinitesimal local Lorentz transformations [21,39], and where $\Gamma_{\kappa\mu\nu}$ and $\Gamma^\lambda_{\mu\nu} = g^{\kappa\lambda}\Gamma_{\kappa\mu\nu}$ are Christoffel symbols of the first and second kind, respectively. In our case, the only nonzero Christoffel symbols are

$$\Gamma_{003} = \Gamma_{030} = -\Gamma_{300} = VV', \qquad \Gamma_{033} = -V', \tag{94}$$

with $V' = \partial V/\partial z$ and

$$\Gamma^\mu_{0\nu} = VV'\begin{pmatrix} V & -1 \\ V^2 - 1 & -V \end{pmatrix}, \qquad \Gamma^\mu_{3\nu} = V'\begin{pmatrix} -V & 1 \\ -V^2 & V \end{pmatrix}, \tag{95}$$

where $\mu, \nu = 0, 3$. After some tedious algebra, the spin connection becomes

$$\Omega_\mu = \frac{1}{4}[\gamma^3, \gamma^0]\left(\delta^3_\mu - V\delta^0_\mu\right)V', \tag{96}$$

such that

$$e^\mu_{\ a}\gamma^a \Omega_\mu = \gamma^a\left(e^0_{\ a}\Omega_0 + e^3_{\ a}\Omega_3\right) \tag{97}$$

$$= \gamma^0\left(\Omega_0 + V\Omega_3\right) + \gamma^3 \Omega_3 = \gamma^3 \Omega_3 = \frac{1}{2}\gamma^0 V'. \tag{98}$$

In our specific case, we have $\alpha^i = \gamma^i \gamma^0 + V\delta^i_3$ and $\Upsilon = V'/2$. In the Weyl representation, we have

$$\gamma^i \gamma^0 = \begin{pmatrix} \sigma^i & 0 \\ 0 & -\sigma^i \end{pmatrix}, \tag{99}$$

with $\sigma^i$ the Pauli matrices and we obtain the Weyl Hamiltonian

$$\mathcal{H}_\chi = -i\left(\chi\boldsymbol{\sigma} + V\boldsymbol{e}_z\right)\cdot\nabla - iV'/2. \tag{100}$$

# B  Lattice model

To solve the scattering problem numerically, we perform a lattice simulation with the KWANT Python package [24]. To this end, we discretize the continuum Hamiltonian (14) giving a one-dimensional chain along the $z$ direction with lattice constant $a$ (in units of $1/2k_0$) and two orbitals per cell. The dimensionless lattice Hamiltonian becomes

$$\hat{H} = \sum_{\boldsymbol{k}_\perp} \sum_n \hat{c}^\dagger_{\boldsymbol{k}_\perp n} \left[ \left( k_\perp^2 - \frac{1}{4} + \frac{2}{a^2} \right) \sigma_x + k_y \, \sigma_y \right] \hat{c}_{\boldsymbol{k}_\perp n} \tag{101}$$

$$+ \left\{ \hat{c}^\dagger_{\boldsymbol{k}_\perp n+1} \left[ \frac{1}{2ia} \left( V_{n+1/2} \, \sigma_0 + \sigma_z \right) - \frac{1}{a^2} \, \sigma_x \right] \hat{c}_{\boldsymbol{k}_\perp n} + \text{h.c.} \right\}, \tag{102}$$

where $n = 0, 1, \ldots, N-1$ labels the cells of the chain and

$$\hat{c}_{\boldsymbol{k}_\perp n} = \frac{1}{\sqrt{N}} \sum_{k_z} e^{ik_z na} \hat{c}_{\boldsymbol{k}}, \tag{103}$$

where $\hat{c}_{\boldsymbol{k}_\perp n} = \left( \hat{c}_{\boldsymbol{k}_\perp n1}, \, \hat{c}_{\boldsymbol{k}_\perp n2} \right)^t$ and $V_n = V(na)$ is the value of the tilt at site $n$. Here, we take the average value of the tilt for hopping between sites $n$ and $n+1$. For all calculations, we take a scattering region of length $L = a(N-1)$ with $N = 201$ that is sufficiently long so that the tilt profile is practically constant at the boundaries which are connected to semi-infinite leads. In our simulations, we considered three different tilt profiles, all with linear horizons. Firstly, we used

$$V(z) = A + B \tanh(Cz + D), \tag{104}$$

with

$$A = \frac{V_R + V_L}{2}, \qquad C = \frac{\alpha}{2} \frac{V_R - V_L}{(V_0 - V_L)(V_R - V_0)}, \tag{105}$$

$$B = \frac{V_R - V_L}{2}, \qquad D = \frac{1}{2} \ln \left( \frac{V_0 - V_L}{V_R - V_0} \right), \tag{106}$$

where $V(0) = V_0$ and $V'(0) = \alpha$ and which is valid for real $D$, i.e., either $V_L < V_0$ and $V_R > V_0$ or $V_L > V_0$ and $V_R < V_0$. Here, $V_0 = \pm 1$ for a black hole and white hole horizon at the origin, respectively and $V_{R,L} = \lim_{z \to \pm\infty} V(z)$. Note that this tilt profile is symmetric about the origin for $V_R + V_L = 2V_0$. Secondly, we considered the tilt profile $V(z) = 1 + (2/\pi) \arctan(\alpha z \pi/2)$ for the case $V_L = 0$ and $V_R = 2$. For this profile, the asymptotic values are approached linearly instead of exponentially. Finally, we also looked at a purely linear tilt profile, given by

$$V(z) = \begin{cases} V_L & z \le 0, \\ V_L + \alpha z & 0 < z < L_0, \\ V_R & z \ge L_0, \end{cases} \tag{107}$$

with $L_0 = (V_R - V_L)/\alpha$.

# C  Scattering matrix for sharp horizon

Here, we give the complete $S$ matrix at low energies for the sharp black hole horizon where we take $V_L = 0$ and $V_R = V$. Expressions exist for the general case but they are too unwieldy. Here, we obtained the low-energy expressions by expanding the spinors, wavevectors, and group velocity up to third order in $\omega$.

When the incoming mode comes from the type-I region ($v$ mode), the scattering coefficients up to second order in $\omega$ are found to be given by

$$r_{uv} = i\frac{2-V}{2+V} - \frac{V^4 - 5V^2 + 8}{(2+V)^2(V^2-1)}2\omega \tag{108}$$

$$+ \frac{V^7 + 3V^6 - 4V^5 - 18V^4 + 5V^3 + 43V^2 + 6V - 32}{(2+V)^3(V^2-1)^2}2i\omega^2\,, \tag{109}$$

$$t_{uv} = i\frac{2\sqrt{V-1}}{2+V} + \frac{V(V-1)(3V+8) - 16}{(2+V)^2(1+V)\sqrt{V-1}}\omega \tag{110}$$

$$- \frac{7V^7 + 27V^6 - 17V^5 - 159V^4 + 6V^3 + 432V^2 + 176V - 256}{4\sqrt{V-1}(2+V)^3(V^2-1)^2}i\omega^2\,, \tag{111}$$

$$t_{vv} = \frac{2\sqrt{1+V}}{2+V} - \frac{iV^2\sqrt{1+V}}{(2+V)^2(V-1)}\omega + \frac{V^5 + 3V^4 - 7V^3 - 7V^2 + 58V + 80}{4(2+V)^3(1+V)^{5/2}(V-1)^2}V^2\omega^2\,, \tag{112}$$

such that whenever the zeroth-order term is real (imaginary), the first-order term is imaginary (real). Hence, the first-order term enters only as a phase such that the transmission probability is constant up to first order. Numerically, we find that this holds at all odd orders such that the transmission functions $R_{u \leftarrow v}$, $T_{u \leftarrow v}$, and $T_{v \leftarrow v}$ are even functions of the energy.

On the other hand, when the incoming mode comes from the type-II region, there are two possibilities: $w_1$ and $w_2$. Up to first order in $\omega$, we find

$$t_{uw_{1,2}} = \sqrt{2}\frac{V^2 + V - 2 \pm i\sqrt{V^2-1}(2-V)}{V(2+V)\sqrt{V-1}} \tag{113}$$

$$\pm \frac{P_1(V) \mp i\sqrt{V^2-1}\left(16 + 12V - 16V^2 - 7V^3 + 3V^4 + V^5\right)}{\sqrt{2}V(1+V)^{3/2}(V^2+V-2)^2}\omega\,, \tag{114}$$

$$r_{uw_{1,2}} = \frac{V^2 - 4 \pm 4i\sqrt{V^2-1}}{\sqrt{2}V(2+V)} \tag{115}$$

$$\pm \frac{P_2(V) \mp i\sqrt{V^2-1}\left(32 + 40V - 12V^2 - 20V^3 - 4V^4\right)}{2\sqrt{2}V(2+V)^2(V^2-1)^{3/2}}\omega\,, \tag{116}$$

$$r_{vw_{1,2}} = \mp\frac{V}{\sqrt{2}(2+V)} \tag{117}$$

$$\pm \frac{4 + 8V - 8V^3 - 4V^4 \mp i\sqrt{V^2-1}\left(12 + 6V - 6V^2 - 3V^2\right)}{2\sqrt{2}(V-1)^2(1+V)^2(2+V)^2}iV\omega\,, \tag{118}$$

where $P_1 = 16 + 12V - 28V^2 - 9V^3 + 12V^4 - 3V^6$, and $P_2 = 32 + 40V - 44V^2 - 54V^3 - 6V^4 + 5V^5$. Taking the expressions of the scattering amplitudes up to first order in $\omega$, one can check that $SS^\dagger = S^\dagger S = 1 + \mathcal{O}(\omega^2)$ where $S$ is defined in Eq. (67).

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
