# Peer review of "Artificial event horizons in Weyl semimetal heterostructures and their non-equilibrium signatures"

_SciPost Physics, doi:SciPost Phys. 11, 095 (2021)_

## Round 1 · Referee Report · Anonymous (Referee 1) · 2021-7-7

Strengths

The Hawking radiation from the black hole is a very interesting and fundamental phenomenon. However, the temperature of Hawking radiation from the black hole is extremely small, and there are practically no chances to observe it. The Hawking effect is so fundamental, that it can reproduced in different types of condensed matter, which experience effective gravity. It appears that the chances to experimentally observe the Hawking effect in Weyl materials are much higher than in any other condensed matter systems. That is why the search for the Hawking effect in Weyl materials is becoming the hot topic. The authors offer an interesting recipe for the proper experiment, and who knows, maybe it will finally lead to the observation of Hawking radiation.

Weaknesses

I have one comment to be addressed. The authors wrote:
“First term in Eq. (1) tilts the Weyl cone and breaks the Lorentz covariance of flat spacetime”.

I do not agree. The Lorentz covariance is not broken. This can be seen when the Hamiltonian (1) is written in terms of the tetrads. The description in terms of tetrads is missing in the paper, that is why it is not clear how the “acoustic” metric emerges in Eqs. (2) and (3). In the correct approach, the metric emerges as the bilinear combination of tetrads, g^{\mu\nu}=\eta^{ab}e^\mu_a e^\nu_b.

I suggest to clarify this point.

The next comment is for the authors information, it concerns the references.
The conical Fermi surface emerging in the overtilted region of type II Weyl system was first discussed in Eq.(14) here:
G.E. Volovik and M.A. Zubkov, Emergent Weyl spinors in multi-fermion systems,
Nuclear Physics B 881, 514 (2014),
and the Berry flux in Weyl systems can be found in Eqs.(4) and (5) here:
G.E. Volovik, Zeros in the fermion spectrum in superfluid systems as diabolical points,
JETP Lett. 46, 98 (1987). However, this remark is for information only. I do not insist on citations.

Report

The Hawking radiation from the black hole is a very interesting and fundamental phenomenon. However, the temperature of Hawking radiation from the black hole is extremely small, and there are practically no chances to observe it. The Hawking effect is so fundamental, that it can reproduced in different types of condensed matter, which experience effective gravity. It appears that the chances to experimentally observe the Hawking effect in Weyl materials are much higher than in any other condensed matter systems. That is why the search for the Hawking effect in Weyl materials is becoming the hot topic. The authors offer an interesting recipe for the proper experiment, and who knows, maybe it will finally lead to the observation of Hawking radiation.

That is why I am in favour of publication.

However, I have one comment to be addressed.

The authors wrote:
“First term in Eq. (1) tilts the Weyl cone and breaks the Lorentz covariance of flat spacetime”.

I do not agree. The Lorentz covariance is not broken. This can be seen when the Hamiltonian (1) is written in terms of the tetrads. The description in terms of tetrads is missing in the paper, that is why it is not clear how the “acoustic” metric emerges in Eqs. (2) and (3). In the correct approach, the metric emerges as the bilinear combination of tetrads, g^{\mu\nu}=\eta^{ab}e^\mu_a e^\nu_b. I suggest to clarify this point.

The next comment is for the authors information. It concerns the references.
The conical Fermi surface emerging in the overtilted region of type II Weyl system was first discussed in Eq.(14) here:
G.E. Volovik and M.A. Zubkov, Emergent Weyl spinors in multi-fermion systems,
Nuclear Physics B 881, 514 (2014),
and the Berry flux in Weyl systems can be found in Eqs.(4) and (5) here:
G.E. Volovik, Zeros in the fermion spectrum in superfluid systems as diabolical points,
JETP Lett. 46, 98 (1987).
However, this remark is for information only. I do not insist on citations.

Requested changes

1) The statements on Lorentz covariance should be corrected.

2) The effective metric should be derived from the tetrads.

  • validity: high
  • significance: top
  • originality: top
  • clarity: good
  • formatting: good
  • grammar: perfect

Author:  Christophe De Beule  on 2021-09-13  [id 1752]

(in reply to Report 1 on 2021-07-07)

We thank the Referee for carefully reading our manuscript and for their helpful comments which allowed us to improve the manuscript. Below, we give a point-by-point response to the referees’ comments.

The authors wrote: “First term in Eq. (1) tilts the Weyl cone and breaks the Lorentz covariance of flat spacetime”. I do not agree. The Lorentz covariance is not broken. This can be seen when the Hamiltonian (1) is written in terms of the tetrads. The description in terms of tetrads is missing in the paper, that is why it is not clear how the “acoustic” metric emerges in Eqs. (2) and (3). In the correct approach, the metric emerges as the bilinear combination of tetrads, $g^{\mu\nu}=\eta^{ab}e^\mu_a e^\nu_b$. I suggest to clarify this point.

Our response: We have corrected the statement on Lorentz covariance in the introduction. We now state correctly in Section 2 that Lorentz covariance is instead broken by quadratic momentum terms in the model Hamiltonian for the tilted Weyl semimetal. We have also added the derivation of the metric in terms of the tetrads in the introduction. The details of this calculation, as well as a short pedagogical introduction to tetrads, is given in the new Appendix A.

The next comment is for the authors information. It concerns the references. The conical Fermi surface emerging in the overtilted region of type II Weyl system was first discussed in Eq.(14) here: G.E. Volovik and M.A. Zubkov, Emergent Weyl spinors in multi-fermion systems, Nuclear Physics B 881, 514 (2014), and the Berry flux in Weyl systems can be found in Eqs.(4) and (5) here: G.E. Volovik, Zeros in the fermion spectrum in superfluid systems as diabolical points, JETP Lett. 46, 98 (1987). However, this remark is for information only. I do not insist on citations.

Our response: We thank the Referee for pointing out these earlier works that we missed and we now cite them in the updated manuscript.

---

## Round 1 · Referee Report · Anonymous (Referee 2) · 2021-8-20

Report

The fact that curved-spacetime effects can be realized artificially in condensed matter systems provides a profound link between two opposite pillars of physics that could be very useful. On one hand, condensed matter systems could potentially be used to measure fundamental effects of quantum gravity in table-top experiments. On the other, elements of gravity physics could be used to engineer quantum materials with interesting properties. In this work, the authors explore manifestations of Hawking radiation – the most high-profile effect of quantum mechanics in curved space – in the condensed matter setting of Weyl semimetals (WSMs). They show that Hawking-radiation like effects could be present in heterostructures of type-I and type-II WSMs if they are driven out of equilibrium, and they propose two clever experimental setups to accomplish this.
Their manuscript first provides a nice review of the basics of WSM physics and cites a useful subset of the literature on artificial gravitational effects in condensed matter. Fig.1 provides a very clear explanation of basic WSM physics and their proposed experimental picture. They provide an explicit mapping from the Weyl Hamiltonian to an effective spacetime metric (Eq. 2) and discuss the interpretation of a junction between type-I and type-II WSMs as an effective event horizon. They then introduce a useful minimal model of a WSM; it has two Weyl nodes of opposite chirality and quadratic correction terms regularize certain divergences.
Focusing on the case of momentum normal to the interface, the authors solve the quantum mechanical problem in increasing complexity. First, the constant-tilt case is reviewed, which provides a basis for their solutions to the non-constant case. The variable-tilt case is then studied in two limits: the slowly-varying case and the abrupt-transition case. In each case, the validity of a Hawking-radiation picture is
In the slowly-varying case, the authors employ a WKB approach based on the constant-tilt solution. The WKB solution breaks down at the “event horizon”; this issue is overcome by a clever technique of “gluing” together the WKB solutions on either side of the horizon and an exact solution for a linear tilt in the vicinity of the horizon. The accuracy of this approximation is demonstrated clearly in Fig. 3. Using this approximate solution to the wave equation, they estimate scattering coefficients for a wave incident on the horizon from the “black hole” (type-II) region. They find a scattering probability that is apparently matches the form of Hawking radiation, and an effective Hawking temperature dependent only on the slope of the tilt profile at the horizon.
The case of a rapidly-varying tilt profile is approximately solved by gluing together constant-tilt solutions with appropriate boundary conditions. The approximation’s validity is demonstrated clearly in Fig. 4. In this case, the analogy to Hawking radiation breaks down.
The authors note that the Hawking-radiation-like transmission through the event horizon produces no effect on transport in equilibrium. They propose and analyze two clever schemes for generating non-equilibrium occupations in the Type-II region in order to induce a Hawking current. This is done by either irradiating the Type-II region with circularly polarized light or by coupling to magnetic leads. They finish with a numerical calculation of the resulting differential conductance. In the extreme limits considered earlier analytically, the numerical results agree nicely.
The paper is well-written, highly likely to be correct, and addresses an interesting, genre-linking question with clever techniques. I believe it meets the acceptence criteria and I endorse publication pending adequate response to the two main issues I raise below.

I have two main criticisms for the authors to address.
1. Both solutions for variable-tilt scenarios with event horizons are based on the constant-tilt solution in some sense, which is treated in Sec. 2. In the constant-tilt case, four distinct zero-energy modes are identified (“u”,”v”,”w1”,”w2”); these are used explicitly in the sharply-varying case and analogues of these modes are identified in the WKB approach. For |V|<1 (type-I), the w1 and w2 modes are called “evanescent”. These modes are used in the scattering calculation that gives the Hawking-like transmission, but it is not clear to me that they really exist as low-energy modes in the type-I regime.

More precisely: for a constant-tilt profile, the z-momentum, “k”, is a well-defined quantum number and an observable taking real values. For a k-eigenstate, Eq. 16 gives the energy. For V > 1, there are 4 solutions for zero-energy (u,v,w1,w2). For V < 1, there are only two real values of k giving zero-energy modes (u,v). This scenario is seemingly depicted in Fig. 2. Well-defined k-eigenstates with real k give a complete basis for the Hilbert space, and all meaningful solutions should be superpositions of these states.

However, the w1,w2 states in the type-I regime seemingly play a key role in the scattering calculation. Please explain in what sense these complex-k “evanescent” states exist as meaningful zero-modes in the type-I regime and why they may be used as reference states in the scattering calculation.

2. I believe it is your responsibility to accurately explain to what extent your proposed measurement really corresponds to a “condensed matter measurement of Hawking radiation”, or to what extent that is a claim you are making. (Your paper opens with a discussion of the difficulty inherent in measuring Hawking radiation from a black hole and a history of theoretical foundation to someday measure curved-spacetime quantum effects in condensed matter settings, with clear implication.)

I don’t challenge the Black hole analogy, in the sense that clearly geodesics may enter the type-II region but not exit it. (Assuming it extends infinitely to the right without another type-I region.) I also agree that you have (a) identified a Hamiltonian with a clear curved-spacetime counterpart featuring a black-hole region, (b) approximately solved this Hamiltonian in some limits, and (c) extracted scattering coefficients corresponding to your solution. But in what sense can you argue that this transmission is really “Hawking radiation”? The common physical picture used to explain Hawking radiation relies on spontaneous particle generation at the event horizon, requiring a theory of interacting particles. But your Hamiltonian is purely non-interacting. You mention that there’s no current in equilibrium and you drive the system out of equilibrium to generate a current, but is this really the same mechanism as Hawking radiation? Naively, it’s not so surprising that driving a system out of equilibrium can create unusual current flow. Is all transport across an effective horizon to be regarded as “Hawking”? The fact that for a quickly varying tilt profile, the Hawking radiation analogy “breaks down” seems to imply otherwise, for the Hamiltonian is still described by an effective black hole region and you still generate a current, though not with the desired universal black-body form.

If the experiment you propose is carried out and your results are verified, what would/could this show? I fully believe that you correctly approximated the solution to the 1D scattering problem you are interested in – would the experiment reveal more than whether or not you are correct?

More minor (and hopefully helpful) feedback follows below:
1. Since the paper crosses boundaries between GR and condensed matter, it may be more helpful to condensed matter readers to define some GR terms more carefully. Specifically, two sentences remarking on technical definitions of “black hole” and “hawking radiation” could be helpful. While your condensed matter readers will know these ideas, precise technical definitions are important if your claim is that they are being realized in a condensed matter system.
2. Sections 1,2 suffer organizationally from unnecessary ‘back-and-forth’ between the constant-tilt and variable-tilt Hamiltonians. (First constant-tilt Weyl is introduced. This is promoted to variable tilt in Eq. 4. Eq. 9 returns us to constant. Eqs. 10/11 generalize to variable-tilt, only for the discussion to return immediately to the constant-tilt case by Eqs. 14,15. Variable-tilt only addressed meaningfully in Sec. 3.)
3. The metric derived in Eq. 2 is standard. However, the technically-correct way to derive an effective metric for a fermionic theory is via the tetrad formalism. It gives the same result.
4. The discussion of null trajectories that starts with Eq. 3 and culminates in the statement “the interface between type-I and type-II semimetals can thus be regarded as an artificial event horizon…” is confused in two ways:
a. In discussion of geodesics, the line elements dx and dy are set by hand to 0. However, for general V(z), the geodesic equation corresponding to the metric in Eq. 2, decouples the x and y coordinates completely each is given by linear motion. The assumption doesn’t have any qualitative effect, but as written it is left unclear whether the claim that “the interface is an artificial event horizon” is dependent on this simplifying assumption. (And it isn’t.)
b. I claim that the discussion of the linear-tilt interface is unnecessary. The point made is that a “type-I on the left, type-II on the right” structure is effectively a “black hole” in that geodesics may enter but not exit. This follows simply and directly from Eq. 4: In a V > 1 region, all geodesics flow right (increasing z), in a V < -1 region, all geodesics flow left. In a |V| < 1 region, geodesics flow both left and right. Half of page 4 is spent analyzing a linear-tilt interface to seemingly conclude that the black hole analogy is valid. This leaves the impression that the black hole analogy is somehow derived/discovered via the treatment of the linear profile, which is misleading.
5. Continuing the quote from point 4: “…where the type-I and type-II regions correspond to a flat spacetime and a black hole…”. While the type-I region is exterior to the effective black hole, it does *not* necessarily correspond to “flat” spacetime. (The metric is not Minkowski. If variable V(z) creates curvature, this is likely not qualitatively dependent on |V|<1.)
6. Related to the last point, a comment on the actual physical curvature of the spacetime would be informative.
7. Eq. 8 is presented in the introduction without any qualifying comment that it is not exact.
8. After Eq. 11 is the statement “Here, we also added the first term to restore Hermiticity.” I claim that this statement is misleading. The new term is the “spin connection” term and is explicitly required in the Hamiltonian for a fermion theory on the manifold defined by the metric in Eq. 2. The current text misrepresents it as an ad-hoc correction term lacking deeper meaning.
9. What is the justification for the labelling of the k_{perp} = 0 channel as the “Hawking channel”? (I don’t require a formal answer in the reply, but when reading I was doubtful this nomenclature was justified or necessary.)
10. Fig. 2 appears to be minorly mislabeled. The left figure is stated in the caption to refer to the (0 < V < 1) type-I regime, but is labelled as “V = 0”, corresponding to the fully-untilted regime.
11. After Eq. 15 is the statement “with corresponding eigenvalues \lambda*c(k)”. This is a mistake of some kind. The eigenvalues of these states under H(+/- k_0, 0,k) are given correctly by Eq.16. (The eigenvalues \lambda*c(k) are indeed correct for the cooked-up operator (H/k-V), which is perhaps how you liked to think of the eigenstates?)
12. The discussion in Sec. 2 is confused due to inconsistent units/scaling. In particular, Eq. 12 gives the Hamiltonian in a dimensionless form. In the very next paragraph, we set k_{perp} = (+/- k_0, 0). To actually plug this into Eq. 12 and get the eigenstates in Eqs. 14,15, the reader has to realize that this really means plugging (k_{perp})^2 = 1/4 into Eq. 12. This is minor, but consistency in units/scaling would make the paper more readable.
13. In Eq. 33, many z-dependent functions with a \mu-subscript are introduced but are not formally defined anywhere.
14. Fig. 3 is very useful for validating the approximate solution. However, the figure would be better if it showed the approximate solutions beyond their regime of validity. Would be informative to see how the approximate solutions fail. The current figure is “cherry-picked” to only show the successful regimes. (Though this was surely done in good faith.)
  • validity: -
  • significance: -
  • originality: -
  • clarity: -
  • formatting: -
  • grammar: -

Author:  Christophe De Beule  on 2021-09-13  [id 1753]

(in reply to Report 2 on 2021-08-20)

We thank the Referee for carefully reading our manuscript and for their helpful comments which allowed us to improve the manuscript. Below, we give a point-by-point response to the referees’ comments.

Main criticisms

Both solutions for variable-tilt scenarios with event horizons are based on the constant-tilt solution in some sense, which is treated in Sec.\ 2. In the constant-tilt case, four distinct zero-energy modes are identified (“u”,”v”,”w1”,”w2”); these are used explicitly in the sharply-varying case and analogs of these modes are identified in the WKB approach. For $|V|<1$ (type-I), the w1 and w2 modes are called “evanescent”. These modes are used in the scattering calculation that gives the Hawking-like transmission, but it is not clear to me that they really exist as low-energy modes in the type-I regime.

More precisely: for a constant-tilt profile, the z-momentum, “k”, is a well-defined quantum number and an observable taking real values. For a k-eigenstate, Eq. 16 gives the energy. For $V > 1$, there are 4 solutions for zero-energy (u,v,w1,w2). For $V < 1$, there are only two real values of k giving zero-energy modes (u,v). This scenario is seemingly depicted in Fig. 2. Well-defined k-eigenstates with real k give a complete basis for the Hilbert space, and all meaningful solutions should be superpositions of these states.

However, the w1,w2 states in the type-I regime seemingly play a key role in the scattering calculation. Please explain in what sense these complex-k “evanescent” states exist as meaningful zero-modes in the type-I regime and why they may be used as reference states in the scattering calculation.

Our response: Modes with real momenta correspond to propagating modes and are the only admissible modes for a translational invariant bulk system, as complex $k$ modes have wavefunctions that are not normalizable. Hence, as the Referee stated correctly, for a translational invariant system the real $k$ modes form a complete set of the Hilbert space. However, for systems with broken translation invariance, such as the semi-infinite systems considered in our paper, there can exist normalizable modes that are localized at boundaries or interfaces. These evanescent modes need to be included to obtain a complete set for the Hilbert space of a semi-infinite system. In the case we consider in our work, interface states indeed coexist with bulk modes. In addition, the fact that we need four momenta to solve the scattering problem (possibly including complex ones) is also apparent from the mathematical structure of the problem. That is, we analyze two coupled differential equations of second order. The scattering problem would consequently be overdetermined if we ignored the evanescent modes. Finally, we note that interface bound states are not only a mathematical necessity, but also carry physical significance. While the evanescent modes do not carry any current and only indirectly influence the scattering matrix, they do for example contribute directly to local observables such as the local density of states near the interface.

I believe it is your responsibility to accurately explain to what extent your proposed measurement really corresponds to a “condensed matter measurement of Hawking radiation”, or to what extent that is a claim you are making. (Your paper opens with a discussion of the difficulty inherent in measuring Hawking radiation from a black hole and a history of theoretical foundation to someday measure curved-spacetime quantum effects in condensed matter settings, with clear implication.)

I don’t challenge the Black hole analogy, in the sense that clearly geodesics may enter the type-II region but not exit it. (Assuming it extends infinitely to the right without another type-I region.) I also agree that you have (a) identified a Hamiltonian with a clear curved-spacetime counterpart featuring a black-hole region, (b) approximately solved this Hamiltonian in some limits, and (c) extracted scattering coefficients corresponding to your solution. But in what sense can you argue that this transmission is really “Hawking radiation”? The common physical picture used to explain Hawking radiation relies on spontaneous particle generation at the event horizon, requiring a theory of interacting particles. But your Hamiltonian is purely non-interacting.

Our response: Our setup does not give rise to Hawking radiation, and we clearly state this in our manuscript. Instead, the type-I and type-II heterostructure gives rise to similar physics in a very precise sense that we detail in our paper, albeit for massless fermions instead of massless bosons, due to the causal structure of the effective spacetime. Namely, both a real black hole and the low-energy electronic theory of the heterostructure feature event horizons. In this respect, the community working on black hole analogs has introduced the terminology of "analog Hawking radiation" or "acoustic Hawking radiation" [1]. Furthermore, Hawking radiation does not require an interacting theory. It is sufficient to consider a free quantum field in a spacetime featuring an event horizon. The original calculation of Hawking considered the fate of a counter-propagating excitation that propagates radially outwards to the nominal horizon at the moment the horizon is formed. This solution is no longer stationary in the presence of a horizon, but it can be decomposed into stationary solutions. This corresponds to a superposition of particle and anti-particle modes whose coefficients are identified via analytic continuation. Finally, one calculates the particle number of the outgoing modes in the vacuum of the ingoing modes. See for example Ref. [1] for the case of a free massless complex scalar field in a 1+1 dimensional curved spacetime.

You mention that there’s no current in equilibrium and you drive the system out of equilibrium to generate a current, but is this really the same mechanism as Hawking radiation? Naively, it’s not so surprising that driving a system out of equilibrium can create unusual current flow. Is all transport across an effective horizon to be regarded as “Hawking”? The fact that for a quickly varying tilt profile, the Hawking radiation analogy “breaks down” seems to imply otherwise, for the Hamiltonian is still described by an effective black hole region and you still generate a current, though not with the desired universal black-body form.

Our response: Driving the system out of equilibrium is not the same mechanism as Hawking radiation, but the resulting observables can be explained by an analogous process. As we explained briefly above, Hawking radiation relies on the presence of anti-particle excitations. In the Weyl semimetal, these excitations depopulate the Fermi sea, giving rise to a non-equilibrium state. Furthermore, as we have demonstrated, scattering processes across the type-I and type-II interface can be understood in the same way as Hawking radiation when considered as a tunneling process through an event horizon [2]. Indeed, the key ingredient for analogs of Hawking radiation, is the presence of an event horizon in the causal structure of the effective spacetime.

Our results can of course be interpreted alternatively in terms of nonequilibrium transport in a type-I and type-II Weyl semimetal heterostructure. However, in our work we focus on understanding transport across the heterostructure in terms of the effective spacetime using the language of Hawking radiation. This idea was proposed in the literature by Volovik [3]. We believe that we have explicitly demonstrated the validity of his proposal, but at the same time we have pointed out some subtleties and differences, which are also part of the message we wish to convey to the reader. Namely, the transport across the interface is "Hawking" only in certain limits, as the Referee pointed out, and the transport signatures which we have suggested are far from trivial. Firstly, one requires nonequilibrium, and secondly there is the fact that transverse channels corresponding to oblique incidence do not give rise to a clean Hawking signature. The former is a fundamental limitation, while the latter might be avoided by the application of a magnetic field along the tilting direction, which would freeze out most transverse degrees of freedom. Note that in the proposal by Volovik, a spherical region of a type-II Weyl semimetal embedded in a type-I Weyl semimetal was considered, which would radiate only for a short while after the type-II region was created. This mechanism is similar to what we propose, in the sense that inducing a type-II phase diabatically leads to a nonequilibrium distribution. Relaxation towards equilibrium then proceeds via analog Hawking radiation in the form of electron-hole pairs, consisting in our proposed setup (in the case of normal incidence) of positive-energy $w_1$ modes and the absence of negative-energy $w_2$ modes.

If the experiment you propose is carried out and your results are verified, what would or could this show? I fully believe that you correctly approximated the solution to the 1D scattering problem you are interested in – would the experiment reveal more than whether or not you are correct?

Our response: The transport experiment that we propose would serve as a proof of principle for analog event horizons in fermionic systems. Moreover, the setup that we propose might be used to engineer scattering matrices, e.g., by tuning the tilt profile, which in certain limits give rise to "analog Hawking radiation". To better distinguish the different tilting regimes, one could additionally apply a magnetic field along the tilting direction. This allows one to freeze out all transverse degrees of freedom except for the Hawking channel. In our work, we propose how to detect signatures of the analog Hawking radiation by driving the system out of equilibrium. Such a setup has potential further applications in terms of electron lensing, which can be readily understood through the effective spacetime and associated gravitational lensing analogies.

We have accordingly expanded our discussion on the experimental implications in the summary.

Minor comments

Here, we only give our answer to the Referees' numbered minor comments.

1.The analogy with a Schwarzschild black hole and the tilted Weyl semimetal only holds for a specific set of coordinates, called Gullstrand-Painlevé (GP) coordinates. In particular, the Weyl equation for a tilt profile $V(r) = -\sqrt{r_S/r}$ corresponds exactly to that of a Weyl fermion in a Schwarzschild spacetime in GP coordinates. Moreover, the geodesics correspond to some extent to the semiclassical equations of motion of a Weyl quasiparticle. However, the acoustic metric, while not a GR metric, i.e., it does not solve the Einstein equations, does feature a similar spacetime structure as a GR black hole in the presence of a horizon $V(z=z_h) = \pm1$. This is why we prefer the term "artificial event horizon" in the title of our paper instead of "artificial black holes", although we do make use of this convenient terminology in the text. The analog Hawking radiation has its origin in this correspondence. Namely, in both cases semiclassical trajectories in the spacetime behind the horizon (whether it be an analog horizon or a GR horizon) are disconnected from the rest of spacetime. Analog Hawking radiation can then be understood in terms of tunneling of particles from behind the horizon to the outside region. These particles are characterized by an effective temperature scale that is related to the effective surface gravity of the analog horizon. However, due to current conservation in our fermionic condensed-matter analog, there is no net emission in equilibrium.

We have added a short discussion on the limits of the GR analogy in the introduction after the discussion of the semiclassical trajectories.

2.We thank the Referee for pointing this out. After carefully considering your suggestion, we still prefer the original structure. In the model section, we first discuss the bulk system, and then we consider a tilt profile where we identify the different transverse channels. We then discuss the spectrum and eigenstates of one particular channel for constant tilt. In our opinion, it would be confusing to first discuss the spectrum and eigenstates of the $\mathbf k_\perp = (\pm k_0,0)$ channel without first discussing the scattering problem.

3.In the new manuscript, we derive the metric with the tetrad formalism. Details of this calculation can be found in the new Appendix A.

4.a We considered normal incidence since this is the focus of Sec. 3 and it works best to highlight the analogy with the GR black or white hole. Away from normal incidence, the spectrum is effectively gapped as the transverse momentum acts like a mass term which couples co- and counterpropagating modes. Moreover, while the geodesic equations for the acoustic metric in terms of an affine parameter $s$ are given by,

$$ \frac{d^2x}{ds^2} = \frac{d^2y}{ds^2} = 0, \qquad \frac{d^2z}{ds^2} = - V V' \left( \frac{dz}{ds} - V \frac{dt}{ds} \right)^2 + V V' \left( \frac{dt}{ds} \right)^2, \qquad \frac{d^2t}{ds^2} = - V' \left( \frac{dz}{ds} - V \frac{dt}{ds} \right)^2, $$
with $V' = \partial V/\partial z$, one can eliminate the affine parameters $s$,
$$ \frac{dx}{dt} = \frac{\pm k_x}{\sqrt{k_\perp^2 + k_z(t)^2}}, \qquad \frac{dy}{dt} = \frac{\pm k_y}{\sqrt{k_\perp^2 + k_z(t)^2}}, \qquad \frac{dz}{dt} = V(z) \pm \frac{k_z(t)}{\sqrt{k_\perp^2 + k_z(t)^2}}, $$
where $\mathbf k_\perp = (k_x,k_y)$ with $k_x$ and $k_y$ constants of motion and $dk_z/dt = -V' k_z$. Hence, we feel the discussion is more clear by focusing on $k_x=k_y=0$. However, we now emphasize in the updated manuscript that our discussion holds qualitatively for any transverse momentum.

4.b We agree with the Referee and have removed the discussion for the linear tilt profile.

5.We have corrected this in the new version of the manuscript.

6.Since the effective curved spacetime for the tilt profile $V(z)$ is essentially two dimensional, it can be shown that the Riemann curvature tensor is completely determined by the Ricci scalar $R$ and the metric, giving \begin{equation} R_{\mu \nu \rho \sigma} = \frac{R}{2} \left( g_{\mu \rho} g_{\sigma \nu} - g_{\mu \sigma} g_{\rho \nu} \right), \end{equation} which we confirmed by direct computation and where the indices run only over $t$ and $z$ and $R /2 = \left( V' \right)^2 + V V''$ with

$$ g_{\mu \nu} = \begin{pmatrix} V^2 - 1 & -V \ -V & 1 \end{pmatrix}. $$
Likewise the Ricci curvature tensor is given by $R_{\mu \nu} = ( R/ 2 ) g_{\mu \nu}$. The other components of the Riemann and Ricci curvature tensors vanish. An example of the Ricci curvature is shown in Figure 1 of the attachment.

It might be instructive to compare the Ricci tensor defined by the tetrads in our analysis to the case of a Schwarzschild black hole. In Gullstrand-Painlevé coordinates, one then has the line element \begin{equation} ds^2 = - dt^2 + \left( dr + \sqrt{\frac{r_S}{r}} \, dt \right)^2 + r^2 d\Omega^2, \end{equation} such that $V(r) = -\sqrt{r_S/r}$ with $r_S$ the Schwarzschild radius. This problem is truly four-dimensional, and the Ricci curvature vanishes. This is not surprising since in dimension $d>2$, any vacuum solution of the Einstein equations has vanishing Ricci curvature. Indeed, for a vacuum solution, we have \begin{equation} R_{\mu\nu} = ( R / 2) g_{\mu\nu}, \end{equation} and taking the trace of both sides yields $R = (d/2) R$ which has a non-trivial solution only for $d=2$.

Hence, for a tilt profile $V(z)$, the GR analogy breaks down at the level of the Einstein equation. In particular, the analog black hole in the setup that we propose does not satisfy the Einstein equation in four-dimensional spacetime. Instead, it satisfies the Einstein equation in an effective two-dimensional spacetime defined by $z$ and $t$ (the $x$ and $y$ directions are decoupled for the acoustic metric that we consider).

To dwell this last point, we recall that in two-dimensional spacetime, any metric yields a vacuum solution since the Einstein tensor always vanishes, $G_{\mu\nu} = R_{\mu\nu} - ( R / 2) g_{\mu\nu} = 0$ for $d=2$, even when the curvature is nontrivial.. Furthermore, in two dimensions, one cannot define a non-vanishing Weyl curvature tensor, which is related to the fact that all two-dimensional spacetimes are conformally flat [1]. Explicitly, for the acoustic metric one can make the following coordinate transformation $(t,z) \rightarrow (u,v)$ with \begin{equation} u = t - \int^z \frac{dz'}{V(z') + 1}, \qquad v = t - \int^z \frac{dz'}{V(z') - 1}, \end{equation} which results in $ds^2 = \left( 1 - V^2 \right) du dv$ which is related to a flat metric by multiplication of a coordinate-dependent prefactor. However, the ($u,v$) coordinates are ill-defined in the presence of a horizon where $V(z=z_h) = \pm 1$. Hence, the type-I and type-II region are both locally conformally flat, although the total spacetime is topologically distinct from a flat spacetime due to the presence of the horizon.

Finally, we want to point out that for the analog Hawking radiation, the curvature plays no important role as far as we understand. Only the behavior of the tilt $V(z)$ near the horizon matters, which is linear in the case of a Schwarzschild black hole. We have added a small comment on the curvature in the new manuscript.

7.We now state accurately that it is a low-energy result.

8.In the condensed-matter setting, such an approach is common in systems with spatially-varying coefficients in the Hamiltonian. However, we agree that for a Weyl Hamiltonian this term arises naturally from the tetrad formalism and has a geometric interpretation in terms of the spin connection, which is now included in the manuscript. We also give an explicit expression for the spin connection in the new Appendix A.

9.We believe this nomenclature is appropriate since only for this transverse momentum, does the transmission probability assume the thermal distribution at low energies for the slowly-varying tilt profile. Hence, the analog to Hawking radiation only strictly holds in this particular case since for a general transverse momentum the spectrum is effectively gapped. We have now emphasized this in the text. For illustration, we show the transmission probability as a function of $k_x$ with $k_y=0$ in Figure 2 of the attachment. Here the blue lines bound the region with propagating modes and the gray curves bound the region with propagating $w$ modes (at $k_x = 0$ it intersects $\omega = \omega_c$).

10.The caption now states that $0 \leq V < 1$ in the type-I region and the left-hand side of the figure now corresponds to $V = 1/2$.

11.This was a mistake due to a change in notation during the writing of the manuscript, where we indeed used the ``cooked-up operator" that the Referee suggested, and we thank the Referee for pointing it out. We have corrected it in the new manuscript.

12.We have fixed this in the new manuscript. After we introduce the dimensionless units, we no longer use dimensionful units unless explicitly stated.

13.In Eq. 33 of the original manuscript, $k_\mu(z)$ is a solution of the local dispersion relation where $\mu$ labels the solution, e.g., $\mu = { u , w_1 , w_2 }$ for the counterpropagating branch. Similarly, $v_\mu(z) = v_\pm(k_\mu(z))$ is the local group velocity and $\varphi_\mu(z) = \varphi_{k_\mu\pm}(z)$ is the wavefunction, where the sign is $+$ for $\mu = { v }$ and $-$ for $\mu = { u, w_1, w_2}$. We have properly defined these functions in the new version of the paper.

14.We thank the Referee for this suggestion. Note that the WKB solution diverges near the turning point (dashed vertical line) due to the $1/\sqrt{v}$ term. We have added some more data points to show this better on both sides of the horizon. Additionally, the WKB solution is ill-defined at the horizon, which is due to the fact that we used low-energy expressions for the momenta and group velocities to construct the WKB wavefunction. On the other hand, the solution for a linear tilt profile near the origin fits perfectly to the exact solution in the type-I region for the given parameters, while in the type-II region it starts to deviate significantly from the exact solution at some point.

References

[1] S. J. Robertson, The theory of Hawking radiation in laboratory analogs, J. Phys. B: At. Mol. Opt. Phys. 45, 163001 (2012). [2] Maulik K. Parikh and Frank Wilczek, Hawking Radiation As Tunneling, Phys. Rev. Lett. 85, 5042 (2000). [3] G. E. Volovik, Black hole and Hawking radiation by type-II Weyl fermions, JETP Lett. 104, 645 (2016).

Attachment:

Figures.pdf

---

## Round 3 · Referee Report · Anonymous (Referee 2) · 2021-9-27

Report

The authors have given sufficient response to all concerns in my original referee report. I am convinced that their article meets all acceptance criteria for SciPost physics and I recommend publication. (Please see my first report for a more thorough analysis of the scientific content of the article.)

---

## Round 3 · Author Response

Dear Editor,

We would like to resubmit our manuscript entitled "Artificial event horizons in Weyl semimetal heterostructures and their non-equilibrium signatures'' to SciPost Physics.

We thank both Referees for carefully reading our manuscript and for their helpful comments which allowed us to improve the manuscript.
We believe that we have addressed all comments of both Referees an have updated our manuscript accordingly. We hope that it is now ready for publication in SciPost Physics.

Sincerely,

Christophe De Beule, Solofo Groenendijk, Tobias Meng, and Thomas L. Schmidt

---

## Round 3 · List of Changes

1. We have added a derivation of the effective metric with the tetrad formalism in the introduction. Details of the calculation and a small introduction to tetrads can be found in the new Appendix A.
  2. We realized that we made an error in deriving the steady state distribution function in Sec. 4.1. In the steady state $\partial n / \partial t$ vanishes (see new Eq. 75) and the term involving $\dot{\bm r}$ is neglected in the long wavelength limit ($\Omega \ll E_F c/v_F$) while the term involving $\dot{\bm k}$ contains the AC response. In the DC limit, one can also neglect the latter term leading to Eq. 78. However, we stress that this does not affect any of our results. In particular, the non-equilibrium distribution function remains the same, but it is now correctly identified with the second-order DC response.
  3. Several minor changes which are detailed in our reply to the Referees. In particular, we have modified the discussion on the semiclassical trajectories and the black hole analogy in the introduction, as well as an additional paragraph in the conclusion concerning experimental implications.

---

## Editorial Decision

published